# LLM-SRBench: A New Benchmark for Scientific Equation Discovery with Large Language Models

**Parshin Shojaee**[* 1] **Ngoc-Hieu Nguyen**[* 2] **Kazem Meidani**[3 4] **Amir Barati Farimani**[3]
**Khoa D Doan**[2] **Chandan K Reddy**[1]

Website: https://github.com/deep-symbolic-mathematics/llm-srbench

## Abstract

Scientific equation discovery has long been a cornerstone of scientific progress, enabling the derivation of laws governing natural phenomena. Recently, Large Language Models (LLMs) have gained interest for this task due to their potential to leverage embedded scientific knowledge for hypothesis generation. However, it is difficult to assess the true discovery capabilities of these methods because existing benchmarks often use well-known equations. This makes them vulnerable to memorization by LLMs and results in inflated performance metrics that do not reflect genuine discovery. In this paper, we introduce LLM-SRBench, a comprehensive benchmark with 239 challenging problems across four scientific domains specifically designed to evaluate LLM-based scientific equation discovery methods while preventing trivial memorization. Our benchmark comprises two main categories: LSR-Transform, which transforms common physical models into less common mathematical representations to test reasoning beyond memorized forms, and LSR-Synth, which introduces synthetic, discovery-driven problems requiring data-driven reasoning. Through extensive evaluation of several state-of-the-art methods, using both open and closed LLMs, we find that the best-performing system so far achieves only 31.5% symbolic accuracy. These findings highlight the challenges of scientific equation discovery, positioning LLM-SRBench as a valuable resource for future research.

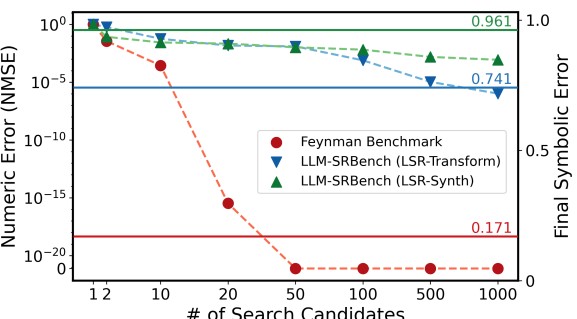

*Figure 1.* Error analysis comparing simple LLM sampling (Llama-3.1-8B) on 100 Feynman problems versus LLM-SRBench datasets (LSR-Transform and LSR-Synth). The sharp drops in numeric error curves and considerably lower symbolic error for Feynman problems suggest memorization rather than gradual discovery.

## 1. Introduction

Equation discovery, the process of uncovering symbolic mathematical expressions from observational data, has been a cornerstone of scientific advancement. This task, also known as symbolic regression (SR), goes beyond mere data-driven predictive modeling by seeking interpretable mathematical relations that reveal the underlying mechanisms of natural phenomena. When scientists derive mathematical equations from empirical data, they gain more than just predictive power – they obtain insights into fundamental physical principles, enable extrapolation beyond observed data, and facilitate knowledge transfer across scientific domains (Langley, 1981; Schmidt & Lipson, 2009).

Standard approaches to equation discovery have primarily relied on genetic programming (GP) and evolutionary algorithms (Cranmer, 2023; La Cava et al., 2021), which represent mathematical expressions as trees and navigate the vast space of possible equations through evolutionary search techniques. However, these methods face two fundamental challenges. First, the NP-hard nature of equation discovery (Virgolin & Pissis, 2022) makes their random mutation and crossover operations computationally prohibitive across

---

[*]Equal contribution [1]Virginia Tech [2]VinUniversity [3]Carnegie Mellon University [4]Capital One. Correspondence to: Parshin Shojaee <parshinshojaee@vt.edu>.

*Proceedings of the 42nd International Conference on Machine Learning*, Vancouver, Canada. PMLR 267, 2025. Copyright 2025 by the author(s).

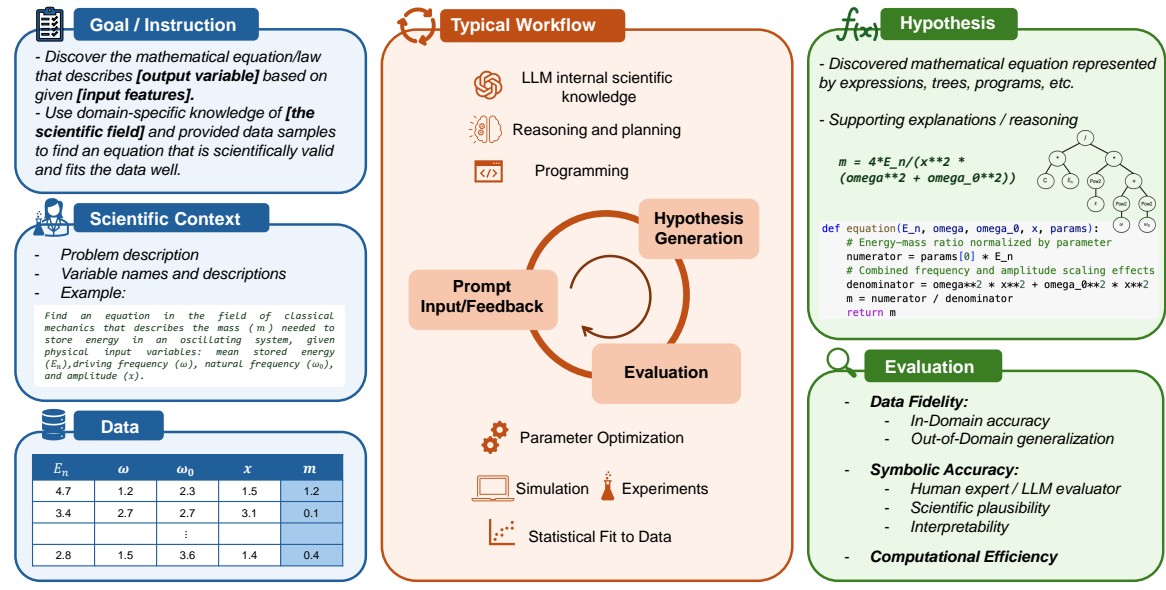

*Figure 2.* **Overview of the LLM-based Scientific Equation Discovery.** The benchmark tasks (**left**) combine scientific context with numerical data. The discovery process (**middle**) iteratively leverages LLM's scientific knowledge and data-driven reasoning to generate hypotheses for underlying equations. Discovered hypotheses, represented as equation strings, trees, or programs, are then evaluated (**right**) using multiple metrics including data fidelity, symbolic accuracy, and computational efficiency.

vast search spaces. Second, unlike human scientists who leverage their domain knowledge and expertise to guide hypothesis formation, these approaches are mostly purely data-driven, and isolated from existing scientific knowledge. These limitations have motivated researchers to develop methods that incorporate scientific domain knowledge into the equation discovery process.

Large Language Models (LLMs) have recently emerged as a promising solution to these challenges, offering a new paradigm for scientific equation discovery. LLMs, trained on vast corpora of scientific literature, possess extensive embedded scientific knowledge. This has sparked significant interest in leveraging LLMs for scientific equation discovery, with several recent works demonstrating their potential (Shojaee et al., 2024b; Ma et al., 2024; Grayeli et al., 2024; Merler et al., 2024; Du et al., 2024; Reddy & Shojaee, 2024; Zhang et al., 2024). These LLM-based approaches have shown to enhance the equation hypothesis generation process by incorporating scientific priors, guiding the exploration of equation search spaces more efficiently, and providing interpretable reasoning for the search process.

Despite the promising potential of LLM-based equation discovery methods, their rigorous and robust evaluation still remains an open challenge. The current scientific equation discovery benchmarks are primarily represented by SRBench (La Cava et al., 2021) and SRSD (Matsubara et al., 2022). SRBench incorporates two key data groups for this purpose: the Feynman physics equations (Udrescu

& Tegmark, 2020), and Strogatz dynamical systems (La Cava et al., 2016; Strogatz, 2018). A notable extension to this framework is SRSD (Matsubara et al., 2022), which enhances the Feynman benchmark by incorporating physically meaningful sampling ranges for data points. However, these benchmarks exhibit significant limitations for the evaluation of LLM-based methods. Their problems are mostly based on known physics equations from textbooks, which makes them often subject to memorization by LLMs.

As noted by (Shojaee et al., 2024b), LLMs frequently succeed on these common equation discovery benchmarks through simple recitation based on variable names and problem descriptions, rather than the actual process of data-driven discovery and reasoning. Our analysis (shown in Fig. 1) also confirms this finding - the sudden drop in the numeric error curve within the first few iterations and significantly lower symbolic error on Feynman problems indicate memorized solutions rather than a meaningful search towards discovery. To mitigate this issue, (Shojaee et al., 2024b; Ma et al., 2024) have introduced a handful of five custom-crafted problems designed to prevent memorization by manually modifying known physical models. While these efforts represent a step forward, the small scale and limited diversity of these problem sets are insufficient to provide a comprehensive evaluation framework for emerging LLM-based methods in scientific equation discovery. A more robust and systematic benchmark is needed to enable standardized evaluation and foster the development of innovative methods in this emerging field.

In this paper, we introduce **LLM-SRBench**, a new benchmark designed to rigorously evaluate the capabilities of LLM-based scientific equation discovery methods. LLM-SRBench addresses the limitations of existing benchmarks by constructing problem sets that avoid trivial recitation while leveraging the scientific priors embedded in LLMs, simulating conditions akin to scientific discovery. The benchmark is structured around two main categories of problems, each targeting distinct aspects of equation discovery. The first category focuses on transforming common scientific problems, such as those from the Feynman equations, into different mathematical representations of the same underlying physical problem. By symbolically altering input-output mappings and generating less common mathematical forms for the same problem, we challenge LLM-based equation discovery to go beyond memorization of the common forms. This approach is motivated by recent findings on the fragility of LLMs' reasoning capabilities to unfamiliar representations of otherwise familiar problems (Mirzadeh et al., 2024; Xie et al., 2024; Wu et al., 2023). The second category extends the approach introduced by (Shojaee et al., 2024b), which combines known terms in the underlying equation with synthetic, novel terms to create problems that go beyond memorization and demand data-driven reasoning. We expand this idea into a comprehensive set of benchmark problems spanning diverse scientific domains. These problems incorporate carefully designed synthetic terms that are both novel and plausible. We further verify the solvability of the generated equations using numerical solvers, ensuring that the benchmark problems remain grounded in physical feasibility while presenting meaningful challenges for LLM-based discovery methods.

LLM-SRBench comprises 111 problems in the first category (LSR-Transform), and 128 problems in the second category (LSR-Synth), spanning four scientific domains: chemistry (36), biology (24), physics (43), and material science (25). We comprehensively benchmark state-of-the-art LLM-based scientific equation discovery methods with several LLM backbones on these datasets. Our experiments reveal several key insights into the capabilities and limitations of current LLM-based scientific equation discovery methods. Results show that the best model can only solve 31.5% of problems on LSR-Transform and 28.1% on LSR-Synth. This underscores the challenging nature of the tasks in LLM-SRBench and highlights its potential as a critical evaluation foundation for future LLM-based scientific equation discovery methods. Overall, the contributions of this work are as follows:

- We introduce **LLM-SRBench**, the first comprehensive benchmark with 239 challenging problems across various scientific domains, designed to evaluate LLM-based scientific equation discovery methods.

- We propose a novel benchmark design through alternative mathematical representations (**LSR-Transform**) and synthetic, discovery-driven problems (**LSR-Synth**) to ensure rigorous evaluation of scientific reasoning and discovery capabilities beyond LLM memorization.

- Extensive experiments on state-of-the-art methods reveal performance peaks at 31%, highlighting the benchmark's challenging nature and its potential for future research.

## 2. LLM-SRBench

We introduce LLM-SRBench, a novel benchmark designed to evaluate LLM-based methods for data-driven scientific equation discovery. As shown in Fig. 2, in this benchmark, a "*data-driven scientific equation discovery*" task is defined as follows: Given a task dataset $\mathcal{D}$, the corresponding scientific context $\mathcal{C}$, the objective is to derive a hypothesis $h$ that represents the underlying mathematical relations behind the data with high precision and scientific plausibility. This process resembles the iterative search and refinement undertaken by human scientists, where LLMs act as optimizers, proposing and refining hypotheses based on both scientific knowledge and empirical data.

### 2.1. LSR-Transform

This category is designed to evaluate whether LLM-based methods can discover equations in less common mathematical forms, avoiding reliance on memorization of well-known representations. This approach is motivated by the observation that LLMs often struggle with unfamiliar instantiations of otherwise familiar problems, as highlighted by recent studies on the fragility of LLM reasoning (Mirzadeh et al., 2024; Xie et al., 2024; Wu et al., 2023). By transforming existing benchmark problems into different mathematical representations, we challenge LLMs' capabilities in data-driven scientific equation discovery and reasoning.

We build on the Feynman (Udrescu & Tegmark, 2020) benchmark (current standard benchmark in scientific equation discovery), which consists of 100 physics equations, and systematically transform these equations into alternative mathematical forms (examples in App. A.1). As demonstrated in Fig. 3(a), the transformation process involves seven key steps: **1) Equation Collection:** We gather the original mathematical expressions, along with their input and output variables, and scientific problem descriptions from the Feynman benchmark. **2) Select Pivot Variable:** For each equation, we choose an input feature to become the new target variable. **3) Feature-Target Transformation:** We transform the dataset by switching the roles of the selected input feature and the original target variable. **4) Symbolic Transformation:** Using the `SymPy` library in Python on the parsed expressions, we solve each equation with respect to the selected input variable, treating it

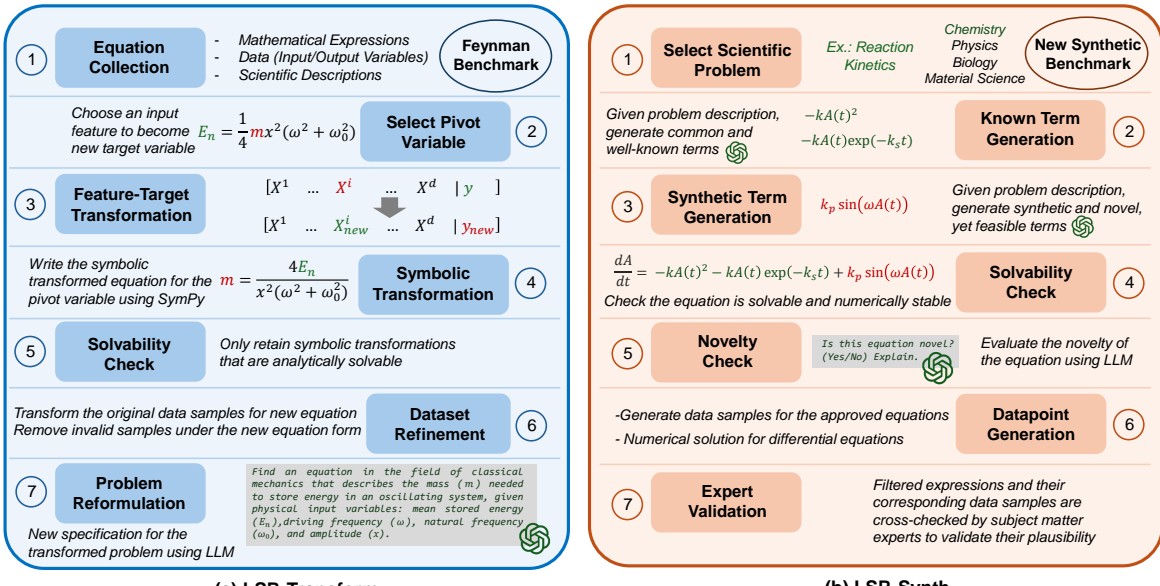

(a) LSR-Transform

(b) LSR-Synth

*Figure 3.* **Data generation pipelines for the two dataset categories in LLM-SRBench.** (a) **LSR-Transform** converts Feynman problems into alternative mathematical forms through symbolic transformation and input-output role switching, and (b) **LSR-Synth** generates novel discovery-driven problems by combining known scientific terms in the underlying models with synthetic novel terms. Both pipelines include validation steps to ensure solvability and scientific plausibility.

as the new output and the original output variable as an input in the transformed equation. **5) Solvability Check:** We retain only those transformations that are analytically solvable, ensuring the feasibility of the resulting equations. **6) Dataset Refinement:** For the transformed equations with altered data domains (e.g., due to square roots or denominators), we filter the original Feynman dataset to ensure all data points fall within the valid domains of the new equations. **7) Problem Reformulation:** Using LLM (GPT-4o), we generate a new natural language specification for each transformed problem. During this data generation process, we constrain the transformed equations' complexity (measured by expression tree node count) to the range of original Feynman dataset distribution (full analysis in Fig. 8, App. A.1). This allows us to focus on the semantic aspects of discovery—specifically the interplay between reasoning and memorization of the mathematical forms—rather than conflating performance with the ability to handle syntactically complex and lengthy hypotheses. We also exclude transformed problems that LLM can solve through direct sampling without requiring access to data.

This process yields 111 total transformed equations derived from the 100 original Feynman problems. Each transformed equation shares the same scientific context, problem description, and variables as its original counterpart but presents a less common mathematical form to be discovered. The goal of LSR-Transform is not to discover new equations but to evaluate whether LLM-based systems can validate discoveries from non-trivial, data-driven transformations of known

equations. To support scientific knowledge-guided discovery, each task in LSR-Transform is supplemented with a natural language description of the scientific problem and dataset, including variable names and their meanings. These descriptions are absent in the original Feynman benchmark but they are needed for LLM-based scientific equation discovery methods to provide scientific context in prompts for knowledge-guided equation discovery by LLMs.

### 2.2. LSR-Synth

This category is designed to assess whether LLMs can discover equations that incorporate new synthetic terms alongside known terms, requiring scientific as well as data-driven reasoning rather than reliance on memorization. The LSR-Synth dataset is motivated by the approach introduced in (Shojaee et al., 2024b) for the handful of manually designed problems and systematically expands it into a comprehensive set of benchmark problems across diverse scientific domains. By combining known terms with synthetic, novel terms, LLMs are challenged to demonstrate discovery capabilities in unobserved contexts, yet leverage their knowledge in the process. The LSR-Synth dataset spans four scientific domains: chemistry, biology, physics, and material science, focusing on key scientific problems, including reaction kinetics in chemistry, population growth in biology, damped harmonic oscillators in physics, and stress-strain relationships in material science (examples in App. A.2).

The data generation process for LSR-Synth involves multi-

ple steps , as illustrated in Fig. 3(b), to ensure the creation of high-quality, challenging benchmark problems: **1) Select Scientific Problem:** We select problems from different scientific domains, such as reaction kinetics in chemistry or population dynamics in biology. **2) Known Term Generation:** Given the problem description, we prompt an LLM (GPT-4o) to generate a list of common and well-known mathematical terms that typically appear in the underlying models. **3) Synthetic Term Generation:** Similarly, we prompt the LLM to generate a list of diverse novel synthetic terms for a given scientific problem, along with descriptions of the problem and variables. For example, in chemistry reaction kinetics, known terms for reaction rate $(dA/dt)$ based on concentration $(A)$ and time $(t)$ might include first-order $(-kA)$ and second-order kinetics $(-kA^2)$ or the exponential decay term $-k\exp(-k_s t)$, while synthetic terms could represent non-linear high-order saturation, e.g., $kA^2/(1+\beta A^4)$, or non-linear quantum tunneling effects, e.g., $kA\exp\left(-\frac{\gamma}{t}\right)/t^2$. **4) Solvability Check:** After sampling from the generated known and synthetic terms and combining them into a complete mathematical expression, we verify the solvability of these expressions using numerical solvers such as `solve_ivp` in Python. This step ensures that the expressions are feasible, providing a basis for generating datapoints. **5) Novelty Check:** In the context of each scientific problem and the complete expression, we evaluate the novelty of the new generated task using LLM (GPT-4o) as a novelty evaluator. This step is to verify that the synthetic terms are novel in the provided context and require data-driven reasoning rather than relying on established knowledge to be discovered. **6) Datapoint Generation:** For expressions that pass the solvability and novelty checks, we generate datapoints using numerical solvers based on the specified initial conditions and parameters. These datapoints are used to create the final task datasets. **7) Expert Validation:** Finally, the filtered expressions, along with visualizations of their generated datapoints, are cross-checked by two subject matter experts to validate their plausibility. After these filtering steps, we finalize a candidate list of 128 problems across the four domains (36: chemistry; 24: biology; 43: physics; and 25: material science). More detailed analysis of LLM-SRBench datasets are provided in App. A.

## 2.3. Evaluation

Evaluating LLM-based scientific equation discovery methods introduces unique challenges due to the open-ended nature of the task and diverse symbolic representation of hypotheses. A discovered equation can be assessed from two perspectives: **(a) data fidelity**, which measures how well the equation fits the observed and out-of-domain (OOD) data, and **(b) symbolic accuracy**, which evaluates the alignment with ground-truth symbolic equation hypotheses. Both

perspectives are critical, as equations may exhibit similar symbolic forms but differ numerically, or vice versa.

**Data Fidelity.** We evaluate data-driven fidelity using two known metrics in equation discovery: (1) Acccuracy to tolerance $\tau$ (Acc$_\tau$) (Kamienny et al., 2022; Biggio et al., 2021), and Normalized Mean Squared Error (*NMSE*). These metrics are computed on both in-domain test data and OOD data (when available) to assess generalization capacity, a crucial requirement for scientific equations.

$$\text{Acc}_\tau = \mathbb{1}\left(\max_{1\le i\le N_{\text{test}}}\left|\frac{\hat{y}_i - y_i}{y_i}\right| \le \tau\right),$$

$$\text{NMSE} = \frac{\sum_{i=1}^{N_{\text{test}}}(\hat{y}_i - y_i)^2}{\sum_{i=1}^{N_{\text{test}}}(y_i - \bar{y})^2}$$

**Symbolic Accuracy.** We evaluate symbolic accuracy with a model-based evaluation strategy using GPT-4o as an evaluator (prompt in App. B, Fig. 11). This approach addresses the limitations of current symbolic metrics like recovery rate in symbolic regression (La Cava et al., 2016), which are very sensitive to exact symbolic matches and fail to account for mathematical equivalence, particularly in different hypothesis representations (e.g., equation as strings, expression trees, or Python programs). Here, GPT-4o evaluates mathematical equivalence by comparing the symbolic form of the predicted hypothesis versus the ground-truth equation after removing parameters and constants. The ability of LLMs to recognize semantic equivalence across different representations makes them particularly well-suited for evaluating LLM-based equation discovery methods, which often operate within a more diverse and open-ended hypothesis space. To validate this metric, two authors also independently evaluated symbolic equivalence on 130 sampled problems, finding 94.6% agreement between GPT-4o and human evaluators. App. B provides more details on the evaluation metrics.

## 3. Experiments

### 3.1. Experimental Setup

We benchmark state-of-the-art LLM-based scientific equation discovery methods using three LLM backbones: one open-source model (`Llama-3.1-8B-Instruct`) and two proprietary models (`GPT-4o-mini` and `GPT-3.5-turbo`). Each discovery task takes as input the problem description, variables, the corresponding dataset, and an instruction specifying the task. The discovery methods then generate and refine equation hypotheses through LLMs. To ensure fair comparison, we standardize each of the methods to use 1k LLM calls per problem while maintaining their core algorithmic designs and hyperparameter settings. Detailed implementation specifics and prompts of each method are provided in App. C. We

*Table 1.* Comparison of different LLM-based scientific equation discovery methods on LLM-SRBench. Performance metrics include symbolic accuracy (SA), numeric precision ($Acc_{0.1}$), and normalized mean squared error (NMSE). Bold values indicate **best performance within each method**, and underlined values show best overall performance across discovery methods.

| Models | LSR-Transform | | | LSR-Synth | | | | | | | | | | | |
| --- | --- | --- | --- | --- | --- | --- | --- | --- | --- | --- | --- | --- | --- | --- | --- |
| | | | | Chemistry | | | Biology | | | Physics | | | Material Science | | |
| | SA (%)↑ | $Acc_{0.1}$(%)↑ | NMSE↓ | SA (%)↑ | $Acc_{0.1}$(%)↑ | NMSE↓ | SA (%)↑ | $Acc_{0.1}$(%)↑ | NMSE↓ | SA (%)↑ | $Acc_{0.1}$(%)↑ | NMSE↓ | SA (%)↑ | $Acc_{0.1}$(%)↑ | NMSE↓ |
| | | | | | | | *Direct Prompting (DataBlind)* | | | | | | | | |
| Llama-3.1-8B-Instruct | 3.61 | 1.801 | 0.3697 | 0.0 | 0.0 | 0.0644 | 0.0 | 0.0 | 0.5481 | 0.0 | 0.0 | 0.0459 | 0.0 | 0.0 | 0.0826 |
| GPT-3.5-turbo | 2.10 | 1.801 | 0.3553 | 0.0 | 8.33 | **0.0023** | 0.0 | 4.16 | 0.5990 | 0.0 | 2.27 | **0.0274** | 0.0 | 0.0 | **0.0277** |
| GPT-4o-mini | **7.21** | **6.306** | **0.2631** | 0.0 | **13.88** | 0.0221 | 0.0 | **4.16** | **0.4648** | 4.54 | **9.09** | 0.0647 | 0.0 | 0.0 | 0.0484 |
| | | | | | | | *SGA (Ma et al., 2024)* | | | | | | | | |
| Llama-3.1-8B-Instruct | 2.70 | 0.909 | 0.3519 | 0.0 | 8.33 | 0.0458 | 0.0 | 0.0 | 0.2416 | 0.0 | 2.27 | 0.1549 | 0.0 | 12.12 | 0.0435 |
| GPT-3.5-turbo | 0.0 | 0.909 | 0.3465 | 0.0 | 8.33 | 0.0071 | 0.0 | 8.33 | 0.1279 | 2.27 | 4.54 | **0.0249** | 0.0 | 28.10 | 0.0019 |
| GPT-4o-mini | **9.91** | **8.11** | **0.2321** | 0.0 | **16.66** | **5.46e-4** | **4.16** | **12.51** | **0.0128** | **4.54** | **9.09** | 0.0511 | 0.0 | **36.11** | **6.02e-4** |
| | | | | | | | *LaSR (Grayeli et al., 2024)* | | | | | | | | |
| Llama-3.1-8B-Instruct | 5.41 | 45.94 | 0.0021 | 0.0 | 27.77 | 2.77e-4 | 4.16 | 16.66 | 2.73e-4 | 4.54 | 25.02 | 0.0018 | 8.21 | 64.22 | 7.44e-5 |
| GPT-3.5-turbo | **12.61** | 47.74 | 0.0015 | 0.0 | 38.89 | 1.51e-4 | 0.0 | 16.66 | 2.31e-4 | 6.81 | 22.71 | 0.0011 | 20.66 | 64.09 | 3.77e-5 |
| GPT-4o-mini | 6.31 | **50.45** | **0.0011** | 2.77 | **38.92** | **9.11e-5** | **8.33** | **20.83** | **1.53e-4** | **9.91** | **31.81** | **9.94e-4** | **28.12** | **72.04** | **9.23e-6** |
| | | | | | | | *LLM-SR (Shojaee et al., 2024b)* | | | | | | | | |
| Llama-3.1-8B-Instruct | 30.63 | 38.55 | 0.0101 | 8.33 | **66.66** | 8.01e-6 | **25.30** | **58.33** | **1.04e-6** | 6.97 | 34.09 | 1.23e-4 | 4.10 | 88.12 | 1.15e-7 |
| GPT-3.5-turbo | 10.81 | 10.81 | 0.1449 | 0.0 | 50.22 | 2.87e-5 | 0.0 | 25.03 | 2.33e-5 | 0.0 | 25.12 | 8.84e-4 | 12.42 | 82.14 | 2.75e-8 |
| GPT-4o-mini | **31.53** | **39.64** | **0.0091** | **11.11** | 52.77 | **4.12e-6** | 16.66 | 29.16 | 3.06e-6 | **9.91** | **36.36** | **7.62e-5** | 20.24 | **88.28** | **3.21e-9** |

evaluate the following discovery methods:

**LLM-SR** (Shojaee et al., 2024b), a program search equation discovery method that generates hypotheses of equation skeleton as Python functions with the main idea of combining LLMs' scientific knowledge with multi-island evolutionary search guided by feedback from data.

**LaSR** (Grayeli et al., 2024), a concept learning equation discovery method that finds abstract textual concepts of mathematical relations from successful equation hypotheses with LLMs and uses these concepts to evolve new hypotheses through a hybrid approach of evolutionary search (with PySR (Cranmer, 2023)) and LLM-guided search.

**SGA** (Ma et al., 2024), a bilevel optimization equation discovery method that iteratively combines LLMs for discrete hypothesis generation of scientific laws and physical simulations in PyTorch for continuous parameter optimization with respect to data.

**Direct Prompting (DataBlind)** serves as a baseline for generating hypotheses purely from contextual information without access to data. By not using data-driven reasoning and refinement in the hypothesis generation, this baseline helps to assess LLMs' memorization of the problem.

### 3.2. Main Results

Our experimental results (Table 1) reveals several key insights into the strengths and limitations of LLM-based scientific equation discovery methods. Overall, performance

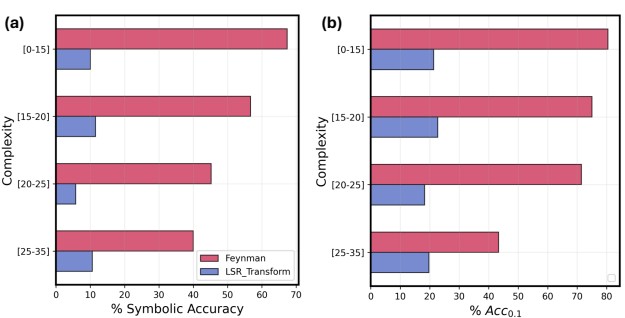

*Figure 4.* **Performance comparison across equation complexity levels for Feynman and LSR-Transform datasets**: **(a)** symbolic accuracy and **(b)** numeric precision ($Acc_{0.1}$) showing considerable performance gap between these two datasets at same complexity levels (averaged over all method-LLM pairs).

remains relatively low across both symbolic and numeric metrics, underscoring the fundamental challenges of this task. One key observation is the poor performance of direct prompting method (DataBlind), which only relies on LLMs' knowledge about the problem without access to data for data-driven refinement. This result underscores the necessity of combining LLM reasoning with observational data, as relying solely on prior knowledge proves insufficient for accurate equation discovery across different problems in LLM-SRBench. We observe that on LSR-Transform data group, LaSR achieves the highest numerical accuracy, leading in both $Acc_{0.1}$ and NMSE, while LLM-SR with GPT-

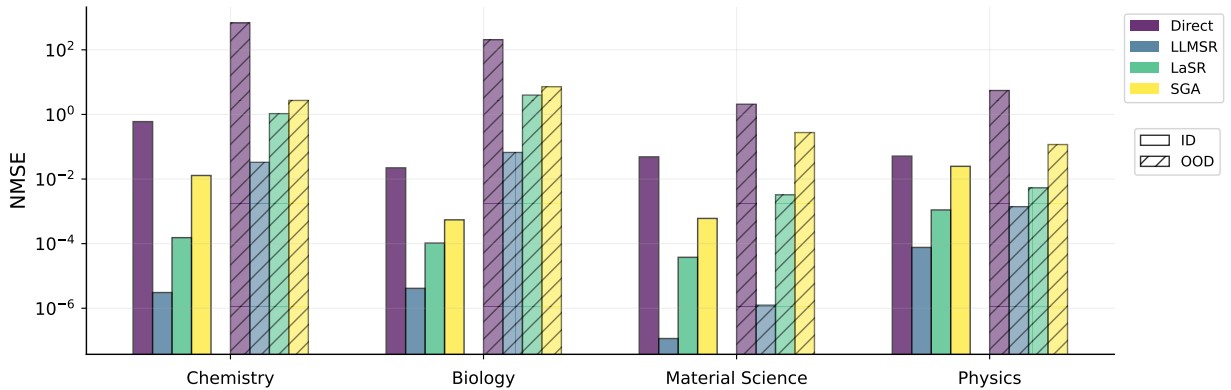

*Figure 5.* Detailed results of in-domain (ID) and out-of-domain (OOD) performance using Normalized Mean Squared Error across various LSR-Synth scientific domains and LLM-based equation discovery methods (with GPT-4o-mini as LLM backbone).

4o-mini outperforms other methods in symbolic accuracy (∼31%). This comparative advantage inverts in the LSR-Synth material science problems, where LaSR consistently yields better symbolic accuracy and LLM-SR achieves better numerical precision, suggesting that different equation discovery strategies may be better suited to different problems.

Another notable observation is the consistent outperformance of models using GPT-4o-mini and Llama-3.1-8B compared to those based on GPT-3.5-turbo. This may be due to improved reasoning architectures or better effectiveness of smaller, less opinionated models in the search and exploration needed for navigating space of possible equations. The lower performance on LSR-Synth compared to LSR-Transform tasks also indicates that the ability to find transformed variants of known problems does not necessarily extend to more challenging scenarios involving novel synthetic terms, where systematic data-driven exploration becomes essential.

### 3.3. Analysis

**LSR-Transform vs. Feynman datasets.** We analyze the performance gap between Feynman and LSR-Transform datasets across different equation complexity levels, measured by the number of nodes in the corresponding expression tree (La Cava et al., 2021). Fig. 4 shows the aggregated average performance (over all methods and LLM backbones) in terms of both symbolic accuracy (a) and numeric precision (b). It can be observed that even at the same complexity levels, LSR-Transform problems are substantially more challenging for current discovery methods than original Feynman problems. Also, this performance disparity persists even for simpler problems ([0-15] nodes), indicating that the challenging nature of LSR-Transform problems for LLM-based scientific equation discovery methods is not necessarily due to the structural complexity.

**Performance on In-domain vs. OOD.** Generalization to unseen data is a fundamental requirement for scientific laws and a critical aspect of equation discovery. A correct mathematical model of observations should not only fit observed data but also extrapolate accurately to out-of-domain (OOD) scenarios. However, current equation discovery benchmarks largely overlook this aspect. In this work, we advocate for explicit OOD assessment in scientific equation discovery by introducing held-out OOD test sets in our benchmark. To systematically evaluate generalization beyond observed data, we generate dedicated OOD test sets for synthetic problems in the LSR-Synth category (see App. A for details on data generation). Fig. 5 provides a comparative analysis of ID vs. OOD results. As expected, all discovery methods exhibit higher NMSE in OOD settings, indicating degraded generalization compared to in-domain data. Among the evaluated methods, LLM-SR achieves the lowest NMSE across both ID and OOD settings, while direct prompting performs the worst. Also, we observe some domain-specific variations in generalization performance: the performance gap between ID and OOD is more pronounced in chemistry and biology problems compared to physics and material science, although the complexity of problems are designed to be similar, as shown in Fig. 10. This suggests that different scientific problems may pose distinct challenges for equation discovery methods, highlighting the need for future research to develop more robust approaches for different scientific disciplines.

**OOD generalization and symbolic accuracy.** We further analyzed the correlation between our proposed symbolic accuracy metric (Sec. 2.3) and data-driven extrapolation performance in OOD settings (averaged over all LSR-Synth domains). As shown in Fig. 6, symbolic accuracy exhibits a strong positive correlation with numerical precision ($\mathrm{Acc}_{0.1}$) on OOD data and a corresponding negative correlation with numerical error (NMSE). This strong correlation observed

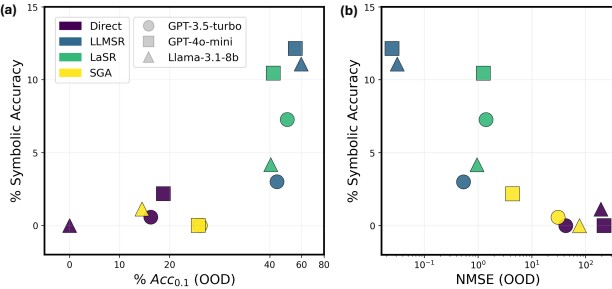

*Figure 6.* **Correlation between symbolic accuracy and OOD performance across different equation discovery methods and LLM backbones**: **(a)** symbolic accuracy vs. $Acc_{0.1}$ showing positive correlation; **(b)** symbolic accuracy vs. normalized mean squared error showing negative correlation. Results are averaged over all LSR-Synth datasets.

between symbolic and OOD performance provides two key insights: First, it establishes OOD evaluation as a powerful approach for assessing the discovery of generalizable equations—an aspect often underutilized in symbolic regression research; second, it validates our LLM-based symbolic evaluation approach through its strong alignment with numeric generalization performance.

More detailed experimental results, including both qualitative analyses of discovered equations and quantitative performance comparisons across scientific equation discovery methods and LLMs, are provided in App. D.

## 4. Related Work

**AI for Scientific Discovery.** Recent advancements in AI for science highlight the ability of LLMs to generate scientific hypotheses by leveraging their extensive knowledge and reasoning capabilities (Lu et al., 2024; Ji et al., 2024; Reddy & Shojaee, 2024). LLM agents, when augmented with external tools and scientific simulators, have shown promise in automated scientific data-driven analysis (Majumder et al., 2024a). While recent benchmarks have been developed to evaluate LLMs and agents in hypothesis generation and scientific question answering (Majumder et al., 2024b; Chen et al., 2024), evaluation for equation discovery and symbolic regression—one of the core tasks in scientific discovery—remains yet unexplored.

**Symbolic Regression.** Symbolic regression approaches fall into three main categories: search-based methods that explore equation spaces via evolutionary algorithms or reinforcement learning (Schmidt & Lipson, 2009; Cranmer, 2023; Petersen et al., 2021; Sun et al., 2023), learning-based methods leveraging pre-trained Transformers on synthetic data (Biggio et al., 2021; Kamienny et al., 2022), and hybrid approaches that guide search using neural priors (Landajuela et al., 2022; Shojaee et al., 2024a; Mundhenk et al., 2021;

Meidani et al., 2023). While these methods have advanced the field of automated symbolic function discovery from data, they mostly lack mechanisms to incorporate scientific domain knowledge into the discovery process.

**LLMs for Equation Discovery.** Recent work has leveraged LLM-based symbolic regression to enhance scientific equation discovery through various approaches leveraging LLMs' knowledge. LLM-SR (Shojaee et al., 2024b) utilizes LLMs' embedded scientific knowledge to generate initial equation hypotheses in the form of Python programming functions, which are then refined through adaptive mutation and crossover operations with LLMs as evolutionary optimizers. In-Context Symbolic Regression (ICSR) (Merler et al., 2024) employs an iterative few-shot learning paradigm over expression candidates, using previously tested successful expressions along with their fitness scores to guide the generation of improved candidates. LaSR (Grayeli et al., 2024) alternates between hypothesis evolution, concept abstraction, and concept iteration phases to build a learned library of scientific concepts for mathematical relations needed to find the equation for a given data. The learned concepts are then used with pure evolutionary search methods (Cranmer, 2023) like PySR (Cranmer, 2023) as well as LLM-guided search to guide the equation hypothesis evolution. Scientific Generative Agent (SGA) (Ma et al., 2024) also implements a bilevel optimization framework for equation discovery where LLMs iteratively propose discrete hypotheses for scientific laws while physical simulations in PyTorch provide experimental validation and data-driven parameter optimization.

**Symbolic Regression Benchmarks.** Symbolic regression benchmarks can be broadly categorized into scientific discovery-oriented and general-purpose mathematical discovery collections. The scientific equation discovery benchmarks are primarily represented by the SRBench (La Cava et al., 2021) and SRSD (Matsubara et al., 2022) benchmarks. SRBench incorporates two key data groups for this purpose: the Feynman physics equations (Udrescu & Tegmark, 2020), and Strogatz dynamical systems (La Cava et al., 2016; Strogatz, 2018). A notable extension to this framework is presented in SRSD (Matsubara et al., 2022), which enhances the Feynman benchmark by incorporating physically meaningful sampling ranges for datapoints. The second category includes benchmarks like the Nguyen collection (Uy et al., 2011) and SRBench's black-box regression problems (La Cava et al., 2016) which include datasets without scientific contexts. However, these existing benchmarks are not well-suited for evaluating LLM-based equation discovery methods. These general-purpose benchmarks focus on the data-driven discovery of abstract mathematical functions without scientific context, while the former scientific benchmarks consist of well-known equations likely memorized by LLMs, enabling success through recitation rather than

scientific reasoning and discovery. Our work extends this line of research by focusing on scientific equation discovery with LLMs, designing the first comprehensive benchmark to assess discovery capabilities of LLM-based scientific equation discovery methods beyond memorization.

## 5. Conclusion

We introduce LLM-SRBench, the first comprehensive benchmark for LLM-driven scientific equation discovery, encompassing 239 tasks across two distinct categories: LSR-Transform (111 problems derived from transformations of established physical models) and LSR-Synth (128 novel synthetic problems spanning four scientific disciplines). Our benchmark provides a standardized and multi-faceted evaluation protocol for assessing scientific equation discovery with LLMs, accommodating diverse hypothesis representations, including expression strings and programs. Extensive experiments with state-of-the-art discovery methods and various LLM backbones on LLM-SRBenchshow a peak performance of only 31%, highlighting the significant challenges and open research opportunities in this domain. We envision that LLM-SRBench benchmark datasets and its evaluation protocol could serve as a foundation for future research, driving progress in automated equation discovery and advancing our understanding of LLMs in symbolic reasoning needed in scientific discovery.

## Impact Statement

The development and future adoption of LLM-SRBench as a benchmark for evaluating LLM-based scientific equation discovery has the potential to significantly impact the field of artificial intelligence for science and scientific discovery. There are many potential societal consequences of our work, none of which we feel must be specifically highlighted here.

### Acknowledgments

This research was partially supported by the U.S. National Science Foundation (NSF) under Grant No. 2416728.

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

# Appendix

# A. Dataset Details

### A.1. LSR-Transform

The LSR-Transform is the first category of datasets in LLM-SRBench, designed to evaluate the ability of LLM-based scientific equation discovery methods in less common mathematical forms. This dataset challenges LLM-based discovery methods to avoid reliance on memorization of well-known representations and instead reason through unfamiliar instantiations of familiar problems. This approach is motivated by the observation that LLMs often struggle with unfamiliar instantiations of otherwise familiar problems, as highlighted by recent studies on the fragility of LLM reasoning (Mirzadeh et al., 2024). By transforming existing benchmark problems into alternative mathematical representations, LSR-Transform provides a rigorous testbed to evaluate how well LLM-based discovery methods perform in both (1) semantic scientific reasoning, which draws on LLMs' built-in scientific knowledge, and (2) data-driven reasoning, which utilizes experimental feedback for equation discovery. LSR-Transform builds on the Feynman benchmark (Udrescu & Tegmark, 2020), a widely used standard benchmark in scientific equation discovery and symbolic regression. The Feynman benchmark consists of 100 physics equations from *Feynman Lecture Series*[1], representing fundamental laws in physics. While the Feynman benchmark has been instrumental in evaluating symbolic regression methods, it primarily tests the ability to recover equations in their standard, well-known forms which are mostly memorized by LLMs. However, real-world scientific equation discovery often involves reasoning about unknown equations based on domain expertise and knowledge from literature as well as empirical data observations. To address this gap, LSR-Transform transforms the original Feynman equations into less common alternative mathematical forms of the same physical problem by switching input-output variables and symbolically solving for the new target variables.

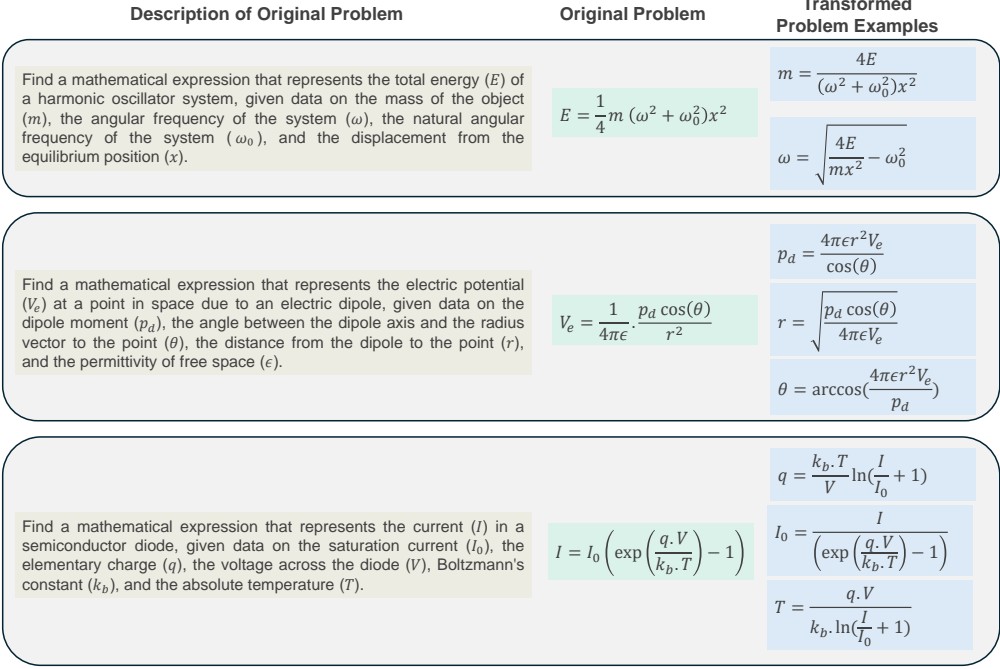

*Figure 7.* Examples of how LLM-SRBench (LSR-Transform) problems can be obtained from original Feynman benchmark problems.

Figure 7 demonstrates the equation transformation process, showing examples of the original Feynman problems (along with their scientific descriptions) and their potential transformed versions. These examples show the dataset's design for altering the mathematical representation of the same problem by analytically solving the equations with respect to different input variables. For instance, the original harmonic oscillator energy equation $E = \frac{1}{4}m(\omega^2 + \omega_0^2)x^2$ is transformed into symbolic representation of $m = \frac{4E}{(\omega^2 + \omega_0^2)x^2}$ and $\omega = \sqrt{\frac{4E}{mx^2} - \omega_0^2}$ where the target variable is switched from energy ($E$)

[1] https://space.mit.edu/home/tegmark/aifeynman.html

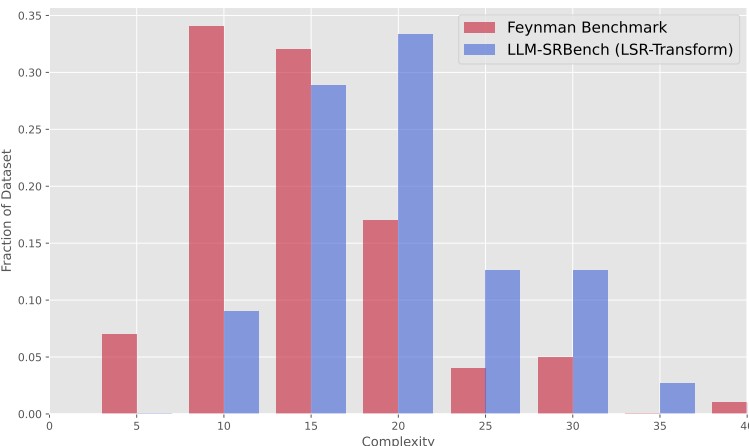

*Figure 8.* Comparison of expression complexity distributions between Feynman Benchmark and LLM-SRBench (LSR-Transform) datasets.

to mass ($m$) or angular frequency ($\omega$). Similarly, in the electric potential equation $V_e = \frac{1}{4\pi\epsilon} \frac{p_d \cos(\theta)}{r^2}$ is also transformed into $p_d = \frac{4\pi\epsilon r^2 V_e}{\cos(\theta)}$, and $r = \sqrt{\frac{p_d \cos(\theta)}{4\pi\epsilon V_e}}$, showcasing how the problem is reformulated to solve for dipole moment ($p_d$), and distance ($r$). These transformations introduce less-common mathematical representations that are simple but not trivial for LLMs to find from the problem description and data. By systematically altering the input-output relationships into new analytically solvable symbolic forms, LSR-Transform challenges models to reason through unfamiliar mathematical forms, testing their ability to generalize beyond memorized representations and leverage data-driven reasoning to find new forms.

The transformed expressions generally exhibit higher complexity than the original physical laws in the Feynman benchmark. To maintain our focus on evaluating semantic complexity (reasoning and memorization capabilities) rather than syntactic complexity and lengthy hypotheses, we deliberately filtered out LSR-transform expressions with significantly higher complexities from the dataset. This filtering ensures that the benchmark primarily challenges discovery models' ability to understand and conduct both scientific and data-driven reasoning rather than their capacity to model longer and more complex mathematical expressions. Figure 8 demonstrates the complexity distribution between the original Feynman Benchmark problems versus their transformed counterparts in LSR-Transform. Following (La Cava et al., 2021), the complexity of each hypothesis (i.e., expression) is quantified as the number of nodes in the expression tree representation of the equation. The expression tree is constructed by parsing the equation into its constituent unary and binary operators, variables, and constants.

Finally, we also exclude the transformed problems that LLM (`Llama-3.1-8B-Instruct`) can solve through direct sampling without requiring access to data. This process creates a dataset of 111 transformed equations, each sharing the same scientific context and variables as its original counterpart but presenting a less common mathematical form. The goal of LSR-Transform is not to discover new equations but to evaluate whether LLM-based systems can guide discoveries from non-trivial, data-driven transformations of known equations.

**Details of Filtering Process** This section provides a comprehensive breakdown of the filtering steps applied during the LSR-Transform dataset generation, addressing the apparent reduction from 100 original Feynman problems to 111 transformed equations. The LSR-Transform dataset generation involves multiple filtering stages that significantly reduce the number of candidate problems. Starting from 100 original Feynman problems, the transformation process initially generates 471 candidate equations by selecting different pivot variables for each equation and performing feature-target transformations. This expansion reflects an average of approximately 4.7 transformed candidates per original problem, demonstrating the diversity introduced by considering multiple input variables as potential targets. The first major filtering occurs during the solvability check using SymPy's symbolic solver (Step 5 in Figure 3), which eliminates 53 problems (11.3% of candidates) that cannot be analytically solved for the target variable. These typically include transcendental equations without closed-form solutions, high-degree polynomial equations where symbolic solutions become intractable, and equations involving complex multi-valued functions. After this stage, 418 problems remain. Notably, no equations are

eliminated during dataset refinement (Step 6 in Figure 3). This stage focuses solely on filtering individual datapoints to ensure they fall within the valid domains of the transformed equations (e.g., ensuring positive values under square roots, avoiding division by zero), while the equations themselves remain intact. The most significant reduction occurs during complexity filtering, where 307 problems (73.4% of remaining candidates) are eliminated, resulting in the final 111 problems. This filtering serves a crucial purpose: to ensure that the challenging nature of LSR-Transform stems from semantic complexity (reasoning about the scientific problem and unfamiliar mathematical forms) rather than syntactic complexity (handling lengthy expressions). Following La Cava et al. (**?**), complexity is measured as the number of nodes in the expression tree representation of each equation. Following this definition, we constrain the complexity distribution to match that of the original Feynman benchmark (Figure 8). In other words, transformed equations with complexity significantly exceeding the original Feynman distribution are exclude. These design choices maintains focus on testing reasoning capabilities while preserving analytical tractability and scientific diversity across physics domains. As demonstrated in Figure 8, even after filtering, LSR-Transform problems remain substantially more challenging than original Feynman problems at same levels of complexity.

## A.2. LSR-Synth

The LSR-Synth is the second category of datasets in LLM-SRBench which is a collection of synthetic problems designed to benchmark the performance of LLMs in scientific equation discovery. This dataset is particularly focused on generating plausible yet challenging equation discovery problems that span multiple scientific domains, including chemistry, physics, biology, and material science. The problems in LSR-Synth are constructed by combining known terms, which are well-established in the scientific literature, with synthetic terms that introduce novel and plausible variations to the equations.

Figure 9 provides examples of problems from the LSR-Synth. These examples demonstrate the dataset's design, which combines well-established mathematical and scientific expressions with novel, domain-specific variations to create challenging models that address the trivial LLM memorization. Each equation is composed of both known and synthetic terms (highlighted in red). Known terms are terms that are commonly found in scientific equations and are well-documented in the literature for that domain and specific problem. For example, terms like $-C_0 A(t)$ and $-C_0 A(t)^2$ are typical in chemistry reactions as the first-order and second-order kinetics. These terms are included to ensure that the problems remain grounded in the established scientific context, providing a foundation for the LLM-based methods to build upon for equation discovery related to each scientific problem. On the other hand, synthetic terms are introduced to create novel variations in the problems to avoid trivial LLM memorization. For instance, terms like $\sin\left(\sqrt{A(t)}\right)$ and $\cos\left(\log\left(A(t)+1\right)\right)$ in chemistry reaction kinetics are designed to challenge the LLM-based discovery models by introducing non-linearities and interactions that are not commonly seen in standard models. These terms are critical for testing the ability of LLM-based equation discovery models to generalize beyond memorization of standard known formulations and discover new patterns from data-driven reasoning and refinement. The combination of known and synthetic terms in LSR-Synth creates a dataset that is both challenging and representative of established scientific problems. This approach enables rigorous evaluation of models' capabilities in interpreting and discovering complex scientific equations, striking a balance between domain familiarity and innovative data-driven reasoning. To generate these known and synthetic terms across various domains, we leverage LLM (GPT-4o) by providing problem domain context and descriptions, prompting it to generate candidate terms. These suggested terms and equations are then filtered based on solvability and novelty criteria, followed by domain expert validation.

Figure 10 provides an analysis of the complexity of the problems in the LSR-Synth dataset. Similar to Figure 8, complexity is quantified as the number of nodes in the expression tree. This figure highlights the diverse nature of the LSR-Synth dataset, with complexity levels ranging from simple expressions to highly complex ones. By spanning a wide range of domains (chemistry, physics, biology, and material science) and hypothesis complexities, LSR-Synth serves as a comprehensive dataset for evaluating the capabilities of LLMs in scientific equation discovery.

Once the structure of equations is generated, their parameters (coefficients) are sampled randomly from specified and scientifically valid ranges, and then data are generated through different solution methods depending on the domain. For dynamical systems (chemical reactions, population dynamics, and physical oscillators), we employ numerical integration using SciPy's `solve_ivp` with the RK45 method, while static relationships (material stress-strain) are evaluated directly over predetermined input ranges. For each domain, we generate 5000 evenly spaced samples. In dynamical systems, these samples span the time interval $t \in [0, 60]$, while for material stress-strain relationships, the samples cover strain $\epsilon \in [0, 0.6]$ and temperature $T \in [273, 573]K$. To evaluate out-of-distribution (OOD) generalization, for time-dependent systems, we designate the last 500 time points as the out-of-domain (OOD) test set, with the remaining 4500 points used for in-domain (ID) training and validation. Similarly, for the stress-strain domain, the OOD test set comprises the last 500 points based

| Example of LSR-Synth Problems with Known and Synthetic Terms |
| --- |

$$-C_0.A(t) - C_1.\sin\left(\sqrt{A(t)}\right)$$

$$-C_0.A(t)^2 - C_1.\exp(-C_2.t) + C_3.\sin(C_4.A(t))$$

$$-C_0.A(t)^2 - C_1.\sqrt{A(t)} - C_2.\cos(\log(A(t)+1))$$

**Chemistry:**
Reaction rate with respect to
Time and Concentration

**Known Terms**
**Synthetic Terms**

$$r\left(1 - \frac{P(t)}{K_0}\right).P(t) + r.\frac{P(t)^2}{\alpha P(t) + 1}$$

$$r.P(t) + r\left(1 - \frac{P(t)}{K_0}\right).P(t) + \beta P(t)\sin(\omega t)$$

$$r\left(1 - \frac{P(t)}{K_0}\right).P(t) + r\left(1 - \frac{P(t)}{K_0}\right).\left(-1 + \frac{P(t)}{\alpha}\right).P(t) + r.(1 - \exp(-\gamma P(t)).P(t)$$

**Biology:**
Growth rate with respect to
Time and Population size

$$C_0.\sin(t) - C_1.x(t) - C_2.x.\exp(-|x(t)|)$$

$$C_0.\sin(t) - C_1.x(t)^3 - C_2.\sin(x(t)).v(t) - C_3.\sin(v(t))$$

$$C_0.\sin(t) - C_1.v(t) - C_2.\sin(x(t)).v(t) + C_3.x(t)^2.v(t) - C_4.x(t).\exp(-|x(t)|)$$

**Physics:**
Acceleration with respect to
Time, Displacement, and Velocity

$$C_0.\left(1 - C_1.(T - T_0)\right).\epsilon + C_2.\exp(-(T - T_0)^2).\epsilon$$

$$C_0.\left(1 - C_1.(T - T_0)\right).\epsilon - C_2(T - T_0) + C_3.(T - T_0).\log(\epsilon + 1)$$

$$C_0.\left(1 - C_1.(T - T_0)\right).\epsilon + C_2.\ \epsilon^{C_3}.\exp(-\frac{C_4}{C_5.T}) + C_6.\exp(-(T - T_0)^2).\epsilon$$

**Material Science:**
Stress with respect to Strain
and Temperature

*Figure 9.* Examples of LLM-SRBench (LSR-Synth) problems with known and synthetic terms across different domains. Each problem presents a target equation as the hypothesis to be discovered which is composed of known terms and synthetic terms (in blue).

on temperature values, maintaining a consistent evaluation framework across all domains. The data generation process incorporates the same quality control criteria used in equation generation. Generated solutions must satisfy: (1) solvability within specified numerical tolerance, (2) meaningful physical behavior (avoiding divergence or constant solutions), and (3) uniqueness compared to existing solutions (using RMSE thresholds). These criteria ensure that the final dataset contains diverse, physically meaningful, and numerically stable solutions suitable for benchmarking equation discovery methods.

## B. Evaluation Details

### B.1. Data Fidelity

We evaluate the data-driven performance of discovered equations through multiple complementary metrics focusing on both predictive accuracy and generalization capability. The primary metrics include Accuracy to Tolerance ($\text{Acc}_\tau$), and Normalized Mean Squared Error (NMSE). The $\text{Acc}_\tau$ metric provides a binary assessment of prediction accuracy based on point-wise relative error. An equation is considered accurate if the maximum relative error across all test tolerance $\tau$. Formally:

$$\text{Acc}_\tau = \mathbb{1}\left(\max_{1 \le i \le N_{\text{test}}} \left|\frac{\hat{y}_i - y_i}{y_i}\right| \le \tau\right)$$

where $\hat{y}_i$ represents the predicted value, $y_i$ is the true value, and $N_{\text{test}}$ is the number of test samples. The indicator function $\mathbb{1}(\cdot)$ returns 1 if the condition is satisfied and 0 otherwise. This metric is particularly useful for cases where maintaining a consistent level of accuracy across all predictions is crucial, as it identifies equations that might have occasional but significant deviations from the true values. NMSE also provides a continuous measure of the overall prediction quality, normalized by the scale of the true values:

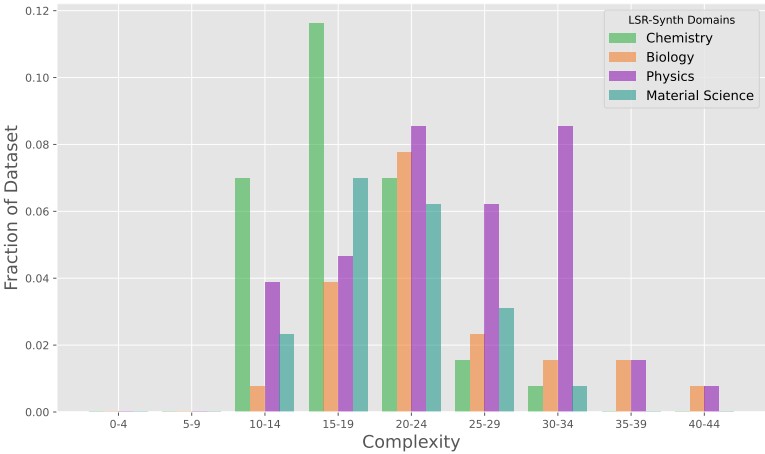

*Figure 10.* Distribution of problem complexity in LLM-SRBench (LSR-Synth) datasets across scientific domains.

$$\text{NMSE} = \frac{\sum_{i=1}^{N_{\text{test}}} (\hat{y}_i - y_i)^2}{\sum_{i=1}^{N_{\text{test}}} (y_i - \bar{y}_i)^2}$$

This normalization makes the metric scale-invariant, allowing meaningful comparisons across different datasets and equation types. The NMSE ranges from 0 to $\infty$, where 0 indicates perfect prediction. Unlike $\text{Acc}_\tau$, NMSE provides a more nuanced view of model performance by considering the magnitude of prediction errors across all test points rather than just their maximum relative error. Beyond standard predictive metrics, we also place particular emphasis on evaluation of out-of-distribution (OOD) generalization, a critical requirement for scientific equations. For datasets in LSR-Synth which have been generated synthetically, we evaluate the discovered hypotheses on held-out OOD test sets to also assess the extrapolation capabilities. The performance gap between in-domain and OOD test sets ($\Delta$NMSE and $\Delta\text{Acc}_\tau$) provides valuable insights into the generalizability of the discovered equations.

### B.2. Symbolic Accuracy

We introduce a novel evaluation methodology for equation discovery that leverages LLM (GPT-4o) as an evaluator for assessing mathematical equivalence between predicted and gold equation hypotheses. Traditional metrics in symbolic regression, such as recovery rate (La Cava et al., 2016), exact match, or normalized tree edit distance (Matsubara et al., 2022), often fail to capture the true semantic equivalence of mathematical expressions, especially when dealing with different representation formats or algebraically equivalent forms. Our approach employs GPT-4o as an automated evaluator, capable of analyzing symbolic equivalence across diverse representation formats including equation strings, expression trees, and executable programs. The evaluation process begins by pre-processing the hypotheses by (1) removing additional information (such as natural language comments in the case of programs), and (2) replacing constants with placeholder parameter vectors, focusing solely on logical structure and mathematical relations. To assess the reliability of this LLM-based symbolic evaluation approach for equation discovery, we conducted a human evaluation study. Two of the authors independently assessed mathematical symbolic equivalence on a set of 130 randomly sampled problems. The validation study revealed a 94.6% agreement rate between GPT-4o and human evaluators, where agreement rate is calculated as the percentage of cases where both LLM and human evaluators made the same judgment about the mathematical equivalence between predicted and ground truth equations (123 out of 130).

Figure 11 provides the prompt used for our GPT-4o based evaluation of the mathematical symbolic equivalence between the generated hypothesis (in the form of program or expression) against the ground truth equation. In this setting, the GPT-4o first articulates its mathematical reasoning before making an equivalence binary assessment.

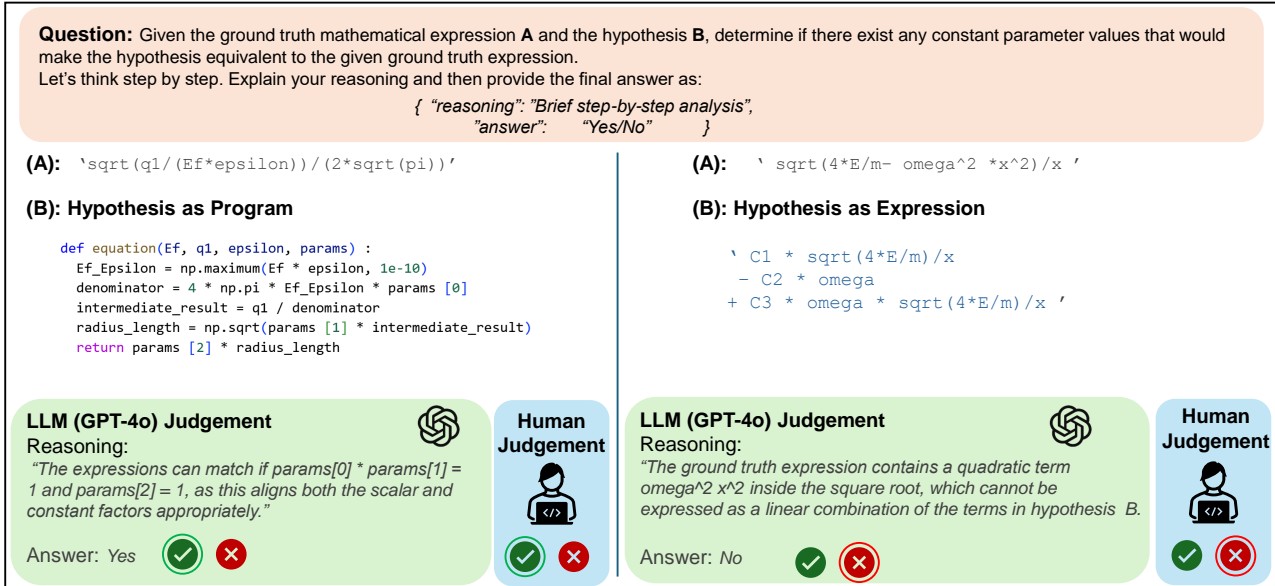

**Question:** Given the ground truth mathematical expression **A** and the hypothesis **B**, determine if there exist any constant parameter values that would make the hypothesis equivalent to the given ground truth expression.
Let's think step by step. Explain your reasoning and then provide the final answer as:

*{ "reasoning": "Brief step-by-step analysis",*
*"answer": "Yes/No" }*

**(A):** `sqrt(q1/(Ef*epsilon))/(2*sqrt(pi))`

**(B): Hypothesis as Program**

```
def equation(Ef, q1, epsilon, params) :
    Ef_Epsilon = np.maximum(Ef * epsilon, 1e-10)
    denominator = 4 * np.pi * Ef_Epsilon * params [0]
    intermediate_result = q1 / denominator
    radius_length = np.sqrt(params [1] * intermediate_result)
    return params [2] * radius_length
```

**LLM (GPT-4o) Judgement**
Reasoning:
*"The expressions can match if params[0] * params[1] = 1 and params[2] = 1, as this aligns both the scalar and constant factors appropriately."*

Answer: *Yes* ✅ ❌

**Human Judgement**
✅ ❌

**(A):** `sqrt(4*E/m- omega^2 *x^2)/x`

**(B): Hypothesis as Expression**

```
C1 * sqrt(4*E/m)/x
 - C2 * omega
 + C3 * omega * sqrt(4*E/m)/x
```

**LLM (GPT-4o) Judgement**
Reasoning:
*"The ground truth expression contains a quadratic term omega^2 x^2 inside the square root, which cannot be expressed as a linear combination of the terms in hypothesis B.*

Answer: *No* ✅ ❌

**Human Judgement**
✅ ❌

*Figure 11.* Symbolic assessment in equation discovery with GPT-4o as evaluator

## C. Implementation Details

For a comprehensive evaluation, we implement four state-of-the-art LLM-guided scientific equation discovery baselines, each tested on LLM-SRBench datasets with three different LLM backbones: an open-source model (`Llama-3.1-8B-Instruct`) and two closed-source models (`GPT-3.5-turbo` and `GPT-4o-mini`).

### C.1. Parameters

Table 2 presents the key implementation details for each discovery agentic method. We adopt most of the hyperparameters from the original implementation for these methods. We have only changed some hyperparameters in different baselines that affect the number of LLM calls in the search framework. This is to make sure we have a fair comparison across baseline discovery frameworks with same access budget to LLMs. In our experiments, all baseline frameworks have 1k calls to LLMs (per problem) through the discovery process.

### C.2. Prompts

#### C.2.1. LLM-SR

We use the default prompts from LLM-SR's (Shojaee et al., 2024b) public code repo (https://github.com/deep-symbolic-mathematics/LLM-SR), which includes:

1. Instruction prompt.

```
You are a helpful assistant tasked with discovering mathematical function structures for scientific systems. Complete
the 'equation' function below, considering the physical meaning and relationships of inputs.
```

2. Evaluation specification prompt.

```
import numpy as np

#Initialize parameters
MAX_NPARAMS = 10
params = [1.0]*MAX_NPARAMS

def evaluate(data: dict) -> float:
    """ Evaluate the equation on data observations."""
```

*Table 2.* Implementation details of LLM-based scientific equation discovery methods.

| Method | Parameters |
|---|---|
| Direct Prompting (DataBlind) | Temperature $\tau = 0.8$
5 equation program hypotheses sampled from LLM for initial prompt
No access to data for data-driven refinement
Time limit $T = 30$s per program hypothesis execution,
BFGS optimizer from Scipy for parameter optimization of equation skeletons |
| SGA (Ma et al., 2024) | PyTorch-based implementation of model and `torch.nn.Module` class
Mean square error loss for data-driven feedback in agentic search
Adam optimizer in PyTorch for differential parameter optimization of equation skeletons |
| LaSR (Grayeli et al., 2024) | Iterations = 25
Cycles per iteration = 550
Populations = 10
Population size = 33
Maximum size = 30
Operators: $+, *, -, /, \wedge$, exp, log, sqrt, sin, cos, tan, cosh
LLM weights: llm_mutate =0.005, llm_crossover =0.005, llm_gen_random =0.005
Top-$K = 20$ concepts from library
Default configuration of PySR for parameter optimization |
| LLM-SR (Shojaee et al., 2024b) | Temperature $\tau = 0.8$
Batch size $b = 4$ equation programs per prompt
$e = 4$ parallel evaluators
Time limit $T = 30$s per program hypothesis,
Memory limit M = 2GB
$m = 10$ islands for population diversity through search
$k = 2$ in-context examples per prompt
Maximum 10 parameters per equation skeleton
BFGS optimizer from Scipy for parameter optimization of equation skeletons |

```
# Load data observations
inputs, outputs = data['inputs'], data['outputs']
X = inputs

# Optimize parameters based on data
from scipy.optimize import minimize
def loss(params):
    y_pred = equation(*X, params)
    return np.mean((y_pred - outputs) ** 2)

loss_partial = lambda params: loss(params)
result = minimize(loss_partial, [1.0]*MAX_NPARAMS, method='BFGS')

# Return evaluation score
optimized_params = result.x
loss = result.fun

if np.isnan(loss) or np.isinf(loss):
    return None
else:
    return -loss
```

3. Equation example specification as Python programming function.

```
### Function Examples
def equation_v0($INPUT_VAR[0], ..., $INPUT_VAR[N], params):

    """ Mathematical function for {$OUTPUT_VAR_DESC}
    Args:
    $INPUT_VAR[0]: A numpy array representing observations of {$INPUT_VAR_DESC[0]}.
    ...
    $INPUT_VAR[N]: A numpy array representing observations of {$INPUT_VAR_DESC[N]}.
    params: Array of numeric constants or parameters to be optimized

    Return: A numpy array representing {$OUTPUT_VAR_DES} as the result of applying the mathematical function to the
    inputs.
    """

    # Equation example 1 logic as function body
    ...

def equation_v1($INPUT_VAR[0], ..., $INPUT_VAR[N], params):
    # Equation example 2
    ...

### Function to be completed
```

```
def equation($INPUT_VAR[0], ..., $INPUT_VAR[N], params):
    """ Improvement version of equation_v0 and equation_v1 """
```

## C.2.2. LASR

We use the default prompts from LaSR's (Grayeli et al., 2024) public code repository ([https://github.com/trishullab/LibraryAugmentedSymbolicRegression.jl](https://github.com/trishullab/LibraryAugmentedSymbolicRegression.jl)), which includes:

1. The LLMINIT prompt, which is used in an LLM-augmented initialization operation.

2. LLMMUTATION prompt is used to mutate an expression based on a set of concepts.

3. LLMCROSSOVER prompt is used to construct a new expression from the crossover of two sampled expressions based on a set of concepts.

4. LLM Concept Abstraction prompt in CONCEPTABSTRACTION function, which extracts a natural language concept from current trends of hypotheses at each iteration.

5. LLM Concept Evolution prompt in CONCEPTEVOLUTION function, which creates a new concept that follows a set of ideas in the current library.

In the following, we provide examples of these prompts.

1. LLMINIT prompt.

```
<System prompt>
You are a helpful assistant that proposes a mathematical expression by following three provided suggestions.
An expression must consist of the following variables: {{variables}}. All constants will be represented with the symbol
C. Each expression will only use these operators: {{operators}}.

<User prompt>
Suggestion 1: {{assump1}}
Suggestion 2: {{assump2}}
Suggestion 3: {{assump3}}

Propose {{N}} expressions that would be appropriate given the suggestions. Provide short commentary for each of your
decisions. End with a JSON list that enumerates the proposed expressions following this format:
```json
["expr1",
 "expr2",
 ...
 "expr{{N}}"
]
```
```

2. LLMMUTATION prompt.

```
<System prompt>
You are a helpful assistant that mutates a mathematical expression by following a few provided suggestions. You will be
given three suggestions and a single reference expression to mutate.
An expression must consist of the following variables: {{variables}}. All constants will be represented with the symbol
C. Each expression will only use these operators: {{operators}}.

<User prompt>
Suggestion 1: {{assump1}}
Suggestion 2: {{assump2}}
Suggestion 3: {{assump3}}
Reference Expression: {{expr}}

Propose {{N}} expressions that would be appropriate given the suggestions and references. Provide short commentary for
each of your decisions. End with a JSON list that enumerates the proposed expressions following this format:
```json
["expr1",
 "expr2",
 ...
 "expr{{N}}"
]
```
```

3. LLMCROSSOVER prompt.

```
<System prompt>
You are a helpful assistant that recombines two mathematical expressions by following a few provided suggestions. You
will be given three suggestions and two reference expressions to recombine.
An expression must consist of the following variables: {{variables}}. All constants will be represented with the symbol
C. Each expression will only use these operators: {{operators}}.
```

```
<User prompt>
Suggestion 1: {{assump1}}
Suggestion 2: {{assump2}}
Suggestion 3: {{assump3}}
Reference Expression 1: {{expr1}}
Reference Expression 2: {{expr2}}

Propose {{N}} expressions that would be appropriate given the suggestions and references. Provide short commentary for
each of your decisions. End with a JSON list that enumerates the proposed expressions following this format:
```json
["expr1",
 "expr2",
 ...
 "expr{{N}}"
]
```
```

### 4. LLM Concept Abstraction prompt.

```
<System prompt>
You are a helpful assistant that hypothesizes about the underlying assumptions that generated a list of good and bad
mathematical expressions in detailed ways. My ultimate goal is to discover what assumptions generated the observed good
mathematical expressions and excludes the bad mathematical expressions. Focus more on the good expressions, their
mathematical structure, and any relation to physical concepts. Note that capital C represents an arbitrary constant

<User prompt>
Good Expression 1: {{gexpr1}}
Good Expression 2: {{gexpr2}}
Good Expression 3: {{gexpr3}}
Good Expression 4: {{gexpr4}}
Good Expression 5: {{gexpr5}}

Bad Expression 1: {{bexpr1}}
Bad Expression 2: {{bexpr2}}
Bad Expression 3: {{bexpr3}}
Bad Expression 4: {{bexpr4}}
Bad Expression 5: {{bexpr5}}

Propose {{N}} hypotheses that would be appropriate given the expressions. Provide short commentary for each of your
decisions. Do not talk about topics related to the simplicity or complexity of the expressions. I want ideas that are
unique and interesting enough to amaze the world's best mathematicians. End with a JSON list that enumerates the
proposed hypotheses following this format:
```json
["hyp1",
 "hyp2",
 ...
 "hyp{{N}}"
]
```
```

### 5. LLM Concept Evolution prompt.

```
<System prompt>
You are an insightful assistant skilled in logical reasoning and deduction. Your task is to analyze a set of ideas and
infer nontrivial conclusions that logically follow from them. The ultimate goal is to uncover underlying principles or
properties of the hidden expressions. Focus on providing logical conclusions that are unique, interesting, and profound.

<User prompt>
Idea 1: {{idea1}}
Idea 2: {{idea2}}
Idea 3: {{idea3}}
Idea 4: {{idea4}}
Idea 5: {{idea5}}

Based on these ideas, deduce {{N}} logical conclusions or hypotheses that directly follow from them. Provide a brief
explanation for each conclusion, highlighting the logical connections between the ideas. Avoid discussing topics related
to the simplicity or complexity of the expressions. Conclude with a JSON list that enumerates the proposed conclusions
in the following format:
```json
["Conclusion 1",
 "Conclusion 2",
 ...
 "Conclusion {{N}}"
]
```
```

### C.2.3. SGA

The following prompts are used in our implementation of SGA (Ma et al., 2024) for scientific equation discovery tasks, following the original implementation SGA's public code repository (https://github.com/PingchuanMa/SGA), which includes:

System prompt for task.

```
You are an intelligent AI assistant for coding and scientific equation discovery.
You are tasked with discovering mathematical function structures for scientific systems.
Follow the user's requirements carefully and make sure you understand them.
Keep your answers short and to the point.
Do not provide any information that is not requested.
Always document your code as comments to explain the reason behind them.
Use Markdown to format your solution.
You are very familiar with Python and PyTorch.
Do not use any external libraries other than the libraries used in the examples.
```

Code formatting prompt for scientific equation discovery task.

```
### PyTorch Tips
1. When working with tensors, always use PyTorch's operators (such as `torch.exp`, `torch.cos`, `torch.sqrt`, ...) to
ensure compatibility and optimal performance.
2. In PyTorch, operator input arguments must be tensors, not floats.

### Code Requirements
1. The only library allowed is PyTorch. Follow the format provided by the user examples.
2. Annotate the size of the tensor as comment after each tensor operation. For example, # (B, 3, 3).
3. Separate the code into parameters that can be tuned with differentiable optimization and the symbolic expression
represented by PyTorch code. Define them respectively in the
5. The proposed code must strictly follow the structure and function signatures below:

```python
import torch
import torch.nn as nn

class SymbolicEquation(nn.Module):

    def __init__(self, {$PARAM_INPUTS}):
        """
        Define trainable continuous parameters for differentiable optimization.
        Tentatively initialize the parameters with the default values in args.

        Args:
            {$PARAM_DESCRIPTION}
        """
        super().__init__()
        {$PARAM_INIT}

    def forward(self, {$INPUT_VARIABLES}) -> torch.Tensor:
        {$FORWARD_FUNCTION_DESCRIPTION$}
```

### Solution Requirements

1. Analyze step-by-step what the potential problem is in the previous iterations based on the feedback. Think about why
the results from previous iterations mismatched with the ground truth. Do not give advice about how to optimize. Focus
on the formulation of the scientific equation. Start this section with "### Analysis". Analyze all iterations
individually, and start the subsection for each iteration with "#### Iteration N", where N stands for the index.
Remember to analyze every iteration in the history.

2. Think step-by-step what you need to do in this iteration. Think about what is needed to improve performance. If the
analysis suggests specific functional forms or constraints, think about how these will be incorporated into the symbolic
equation. Think about how to separate your algorithm into a continuous parameter part and a symbolic expression model
part. Describe your plan in pseudo-code, written out in great detail. Remember to update the default values of the
trainable parameters based on previous optimizations. Start this section with "### Step-by-Step Plan".

3. Output the code in a single code block "```python ... ```" with detailed comments in the code block. Do not add any
trailing comments before or after the code block. Start this section with "### Code".
```

Context prompt for each scientific problem.

```
### Context

The objective is to construct a mathematical expression that accurately maps input variables to a target output based on
a provided dataset. The task involves filling in a code block to define a symbolic expression or model that minimizes
the difference between predicted and ground-truth outputs. The code block defines a class with two functions: one for
parameters within the expression and another for generating or modifying the symbolic structure of the expression.
Feedback is provided in the form of metrics measuring the error between the model's predictions and the ground-truth
values, as well as guidance on structural improvements to the symbolic expression.

The expression represents {$OUTPUT_VAR_DESC}, given data on {$INPUTS_DESC}.
```

# D. Additional Results and Analysis

**Detailed Numeric Accuracy Analysis.** While Table 1 presents median Normalized Mean Squared Error for each method-LLM combination across LLM-SRBench datasets, Figure 12 provides a more comprehensive view of error distributions across all samples. These box plots illustrate performance variations across LLM-SRBench datasets from two perspectives:

comparing different equation discovery methods with GPT-4o-mini as the LLM backbone, and examining different LLM backbones when using LLM-SR method. The substantial variance in NMSE performance across samples reflects the diverse complexity inherent in our benchmark—stemming from both the varying mathematical transformations in LSR-Transform and the different combinations of known and synthetic terms in LSR-Synth datasets. Notably, the relative difficulty of datasets varies across methods and LLM backbones, suggesting that different methods and LLMs possess distinct capabilities in terms of leveraging domain knowledge, reasoning, and generating novel hypotheses.

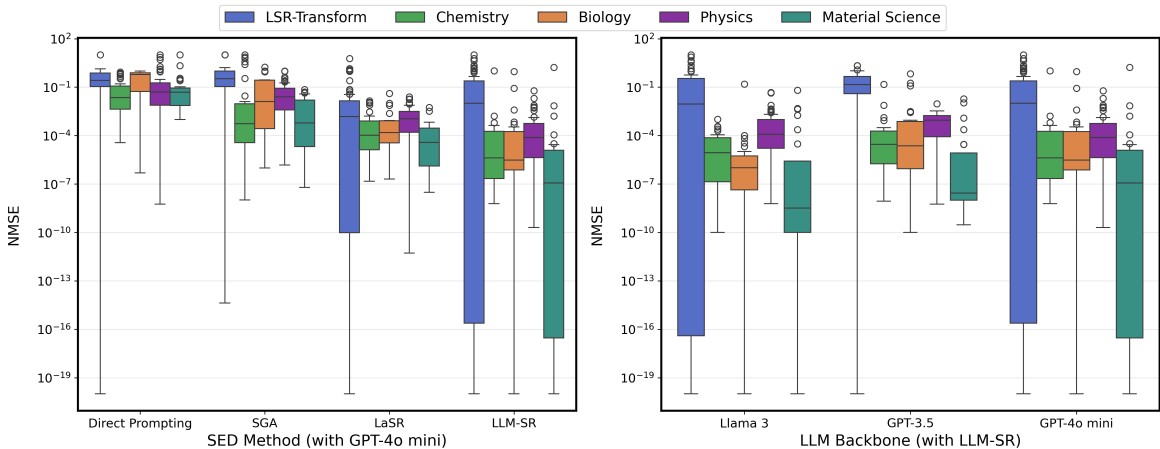

*Figure 12.* Normalized Mean Squared Error (NMSE) of discovered equations in various domains of LLM-SRBench with respect to **(left)** different equation discovery methods using GPT-4omini LLM backbone, and **(right)** different LLM backbones using LLM-SR method

**Symbolic Accuracy and Generalization.** For scientific equation discovery methods, both symbolic accuracy and out-of-domain generalization serve as crucial evaluation metrics, reflecting the methods' ability to uncover true governing equations. Figure 13 examines the relationship between these metrics, plotting symbolic accuracy against both OOD accuracy and OOD NMSE across all method-LLM-domain combinations in LSR-Synth. The strong correlation observed between symbolic and OOD performance yields two important insights: first, it establishes OOD evaluation as a powerful metric for assessing the discovery of generalizable equations, an approach historically underutilized in symbolic regression; second, it validates our LLM-based symbolic evaluation approach through its strong alignment with numeric generalization performance.

**Qualitative Analysis of Outputs.** To provide deeper insights into the behavior of different discovery methods, Figure 14 illustrates their final discovered hypotheses on a biological population growth problem (BPG0) using Llama-3.1-8B as the LLM backbone. Direct Prompting (Figure 14(a)) generates equations that capture basic population dynamics, demonstrating LLMs' ability to propose scientifically plausible structures. SGA's solution (Figure 14(b)) successfully incorporates one of the common population growth terms while exploring additional structural components. LaSR (Figure 14(c)) discovers an equation structure that combines multiple interaction terms, though it differs from established scientific formulations. LLM-SR (Figure 14(d)) combines both standard population dynamics terms and synthetic components in its solution. These examples demonstrate the diverse approaches methods take in balancing scientific interpretability with mathematical expressiveness when discovering equation structures.

## E. Discussion and Future Directions

Our findings from LLM-SRBench reveal several key insights that inform the design of future LLMs for scientific discovery applications. Scientific equation discovery remains a challenging problem for LLMs, requiring a complex interplay of domain knowledge, search capabilities with data-driven feedback, and mathematical manipulation skills. Our results demonstrate that this problem poses significant challenges for LLM-based discovery frameworks across different model architectures, suggesting that current approaches may be fundamentally limited in their ability to perform genuine scientific discovery.

This work questions the current evaluation paradigm for equation discovery in emerging LLM-based techniques. We

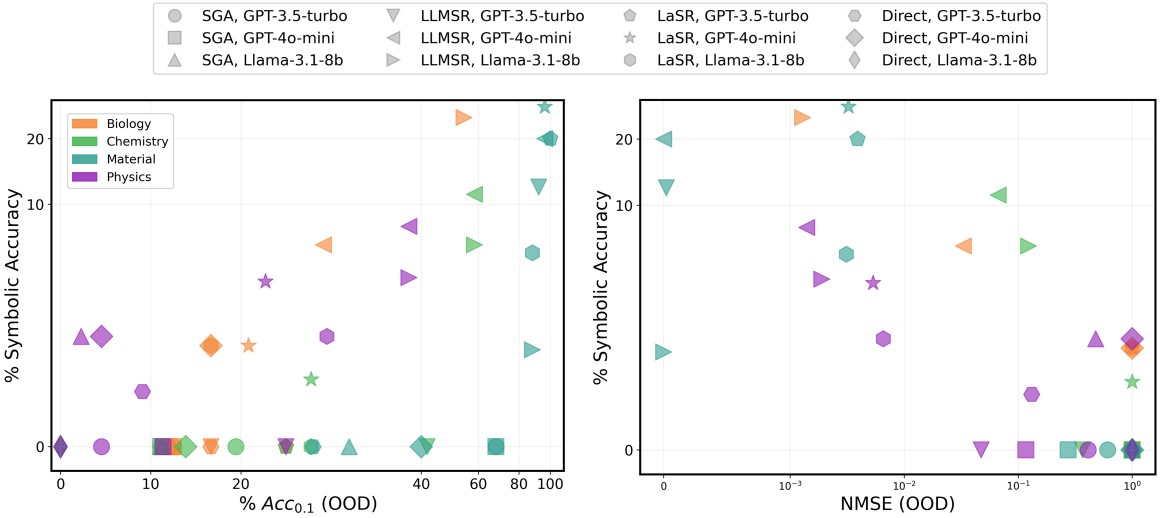

*Figure 13.* Symbolic Accuracy versus OOD performance over all domains, methods, and backbone LLM pairs.

demonstrate that existing benchmarks for this task are susceptible to memorization and inadequate for evaluating these techniques' true scientific discovery capabilities. Motivated by these limitations, we designed LLM-SRBench to address the memorization issue through two key innovations: synthetic imaginary scenarios (LSR-Synth category) that are not based on existing scientific knowledge and require data-driven discovery tools for solution, and transformed equations (LSR-Transform category) that convert common forms of scientifically known equations into less familiar formulations. The LSR-Synth category targets genuine innovation in LLM-based discovery techniques by eliminating the possibility of recalling memorized equations, while LSR-Transform problems are difficult to recite from memory and require reasoning over hypothesis generation steps, making them suitable candidates for evaluating recently emerging LLM-based scientific discovery agents. While the mathematical transformations in LSR-Transform are algebraically valid, their scientific meaningfulness varies considerably across contexts. Many transformations correspond to legitimate physics problems from the Feynman Lecture Series collection and represent alternative problem formulations with practical significance. For example, in the Harmonic Oscillator Energy problem, the original formulation $E = \frac{1}{4}m(\omega^2 + \omega_0^2)x^2$ expresses energy as a function of system parameters, while the transformed version $m = \frac{4E}{(\omega^2 + \omega_0^2)x^2}$ determines the mass required for given energy storage. This transformation maintains scientific meaning by addressing the engineering question of what mass is needed to store a specific amount of energy in an oscillating system, and such inversions are common in engineering design problems where system parameters must be determined to achieve desired performance characteristics. Similarly, the Electric Potential problem transforms from $V_e = \frac{1}{4\pi\epsilon}\frac{p_d\cos(\theta)}{r^2}$ (potential at a point due to a dipole) to $r = \sqrt{\frac{p_d\cos(\theta)}{4\pi\epsilon V_e}}$ (distance for a given potential), addressing the practical question of determining measurement distances in electrostatic experiments or sensor design.

However, not all transformations maintain clear physical interpretability. Some result in equations where the target variable appears in complex functional forms that may not correspond to natural physical questions, such as solving for angular frequency in oscillatory systems yielding expressions involving square roots of differences that lack intuitive physical meaning. Additionally, certain transformations may obscure natural causal relationships—transforming from "force causes acceleration" to "acceleration determines force" maintains mathematical validity but may not reflect underlying physical causality. The LSR-Transform category represents a deliberate balance between mathematical rigor and physical meaningfulness by constraining the complexity of transformed problems to match original problems, focusing on semantic rather than syntactic challenges in scientific equation discovery, while maintaining the original scientific context and variable meanings to ensure that underlying physics remains relevant even when mathematical formulation changes. The varying scientific meaningfulness of transformations reflects broader challenges in automated scientific discovery that warrant future investigation. Automated discovery systems must incorporate mechanisms to evaluate not only data-driven correctness but also scientific plausibility and interpretability of generated hypotheses, as mathematical validity alone is insufficient for meaningful scientific contribution. The most effective approach to scientific equation discovery likely involves close collaboration between AI systems, which excel at exploring vast hypothesis spaces, and human domain scientists, who can

assess scientific meaningfulness and guide discovery directions based on deep contextual understanding. Future equation discovery methods could improve by incorporating literature retrieval tools to build grounding foundations for scientific context and domain knowledge, helping to prioritize discoveries that are mathematically valid, data-consistent, novel, and scientifically meaningful. The field needs evaluation frameworks that assess not just mathematical correctness but also scientific novelty, interpretability, and practical applicability of discovered equations, moving beyond narrow accuracy metrics toward a more comprehensive understanding of what constitutes valuable scientific discovery in the age of LLMs with their vast scientific knowledge.

## F. Comparison with Standard (non-LLM) Symbolic Regression Baselines

To further validate the utility of LLM-SRBench and demonstrate the advantages of LLM-based approaches, we conducted additional experiments comparing LLM-based methods with traditional symbolic regression techniques that do not incorporate domain knowledge. We evaluated PySR (Cranmer, 2023), a state-of-the-art symbolic regression method based on genetic programming, on all LLM-SRBench datasets. PySR operates purely on numerical data points without access to the scientific context, variable descriptions, or domain knowledge that LLM-based methods can leverage in discovery process. We used PySR's default configuration with the same computational budget (equivalent number of evaluations) as the LLM-based methods to ensure fair comparison. Table 3 presents the performance comparison between the best-performing LLM-based method from Table 1 and PySR across all LLM-SRBench datasets. The results reveal several key insights about the complementary strengths and limitations of non-LLM versus LLM-based approaches in equation discovery.

PySR demonstrates competitive and sometimes even better numerical accuracy ($\text{Acc}_{0.1}$) across all datasets. However, PySR consistently shows significantly lower symbolic accuracy, particularly struggling with non-physics domains where it achieves 0% symbolic accuracy on chemistry, biology, and material science datasets. The performance gap is most pronounced in problems that require specialized scientific knowledge. While PySR can fit mathematical patterns in the data, it lacks the scientific intuition to discover equations that align with established physical principles or domain-specific terminology. Interestingly, PySR shows relatively better performance on physics problems, achieving modest symbolic accuracy of 4.54% on LSR-Synth Physics and 8.11% on LSR-Transform (which is based on Feynman physics equations). This suggests that physics problems may contain mathematical patterns that are more aligned with the dictionary design in PySR. So they can be discovered better through the data-driven search pipeline designed in PySR. These findings strengthen the motivation for LLM-based scientific equation discovery and demonstrate that LLM-SRBench successfully captures challenges in equation discovery that traditional symbolic regression methods cannot adequately address through numerical data-driven optimization alone.

*Table 3.* Performance comparison between LLM-based methods and state-of-the-art non-LLM symbolic regression baseline PySR on LLM-SRBench. SA = Symbolic Accuracy (%), $\text{Acc}_{0.1}$ = Accuracy to tolerance 0.1 (%).

| Dataset (Metric) | LLM-SR (best) SA / $\text{Acc}_{0.1}$ | LaSR (best) SA / $\text{Acc}_{0.1}$ | SGA (best) SA / $\text{Acc}_{0.1}$ | PySR SA / $\text{Acc}_{0.1}$ |
|---|---|---|---|---|
| LSR-Transform | 31.53 / 39.64 | 12.61 / 50.45 | 9.91 / 8.11 | 8.11 / 56.76 |
| LSR-Synth Chemistry | 11.11 / 66.66 | 2.77 / 38.92 | 0 / 16.66 | 0 / 41.67 |
| LSR-Synth Biology | 25.30 / 58.33 | 8.33 / 20.83 | 4.16 / 12.51 | 0 / 25.0 |
| LSR-Synth Physics | 9.91 / 36.36 | 9.91 / 31.81 | 4.54 / 9.09 | 4.54 / 29.55 |
| LSR-Synth Material Science | 20.24 / 88.28 | 28.12 / 72.04 | 0 / 36.11 | 0 / 68.0 |

Table 4: LSR-Synth mathematical equations for each scientific domain.

| Domain | Equation ID | Equation |
|---|---|---|
| Chemistry | CKR1 | $-kA(t)^2 + k_z A(t)^2/(\beta A(t)^4 + 1)$ |
| | CKR2 | $-kA(t)^2 - kA(t) + k_w \cos(\log(A(t) + 1))$ |
| | CKR3 | $-kA(t) + k_w \cos(\log(A(t) + 1))$ |

Continued on next page

Table 4 – continued from previous page

| Domain | Equation ID | Equation |
|---|---|---|
| | CKR4 | $-kA(t)^2 - kA(t)\exp(-k_s t) + k_w\cos(\log(A(t)+1))$ |
| | CKR5 | $-kA(t)^2 + k_q A(t)\log(\gamma t + 1)$ |
| | CKR6 | $-k\sqrt{(A(t)} + k_f A(t)^{0.33}$ |
| | CKR7 | $-kA(t)\exp(-k_s t) + k_m\sin(\sqrt{A(t)})$ |
| | CKR8 | $-kA(t)\exp(-k_s t) + k_w\cos(\log(A(t)+1))$ |
| | CKR9 | $-kA(t)^2 - kA(t) + k_t\sin(\log(A(t)+1))$ |
| | CKR10 | $-k\sqrt{A(t)} + k_w\cos(\log(A(t)+1))$ |
| | CKR11 | $-kA(t)^2 + k_t\sin(\log(A(t)+1))$ |
| | CKR12 | $-kA(t)^2 + k_m\sin(\sqrt{A(t)})$ |
| | CKR13 | $-kA(t)\exp(-k_s t) + k_t\sin(\log(A(t)+1))$ |
| | CKR14 | $-kA(t) + k_p\sin(\omega A(t))$ |
| | CKR15 | $-k\sqrt{A(t)} - kA(t)\exp(-k_s t) + k_p\sin(\omega A(t))$ |
| | CKR16 | $-k\sqrt{A(t)} - kA(t)\exp(-k_s t) + k_t\sin(\log(A(t)+1))$ |
| | CKR17 | $-kA(t) + k_f A(t)^{0.33}$ |
| | CKR18 | $-kA(t)\exp(-k_s t) + k_f A(t)^{0.33}$ |
| | CKR19 | $-kA(t)^2 + k_p\sin(\omega A(t))$ |
| | CKR20 | $-kA(t)^2 - kA(t)\exp(-k_s t) + k_t\sin(\log(A(t)+1))$ |
| | CKR21 | $-kA(t)\exp(-k_s t) + k_p\sin(\omega A(t))$ |
| | CKR22 | $-kA(t)\exp(-k_s t) + k_q A(t)\log(\gamma t + 1)$ |
| | CKR23 | $-kA(t)^2 - kA(t)\exp(-k_s t) + k_z A(t)^2/(\beta A(t)^4 + 1)$ |
| | CKR24 | $-k\sqrt{A(t)} + k_p\sin(\omega A(t))$ |
| | CKR25 | $-k\sqrt{A(t)} - kA(t)^2 + k_f A(t)^{0.33}$ |
| | CKR26 | $-kA(t) + k_t\sin(\log(A(t)+1))$ |
| | CKR27 | $-kA(t)^2 - kA(t)\exp(-k_s t) + k_m\sin(\sqrt{A(t)})$ |
| | CKR28 | $-kA(t)^2 - kA(t)\exp(-k_s t) + k_f A(t)^{0.33}$ |
| | CKR29 | $-kA(t)\exp(-k_s t) + k_z A(t)^2/(\beta A(t)^4 + 1)$ |
| | CKR30 | $-kA(t) - kA(t)\exp(-k_s t) + k_z A(t)^2/(\beta A(t)^4 + 1)$ |
| | CKR31 | $-kA(t) - kA(t)\exp(-k_s t) + k_t\sin(\log(A(t)+1))$ |
| | CKR32 | $-k\sqrt{A(t)} - kA(t) + k_w\cos(\log(A(t)+1))$ |
| | CKR33 | $-kA(t) - kA(t)\exp(-k_s t) + k_f A(t)^{0.33}$ |
| | CKR34 | $-k\sqrt{A(t)} - kA(t)^2 + k_t\sin(\log(A(t)+1))$ |
| | CKR35 | $-kA(t)^2 + k_f A(t)^{0.33}$ |
| | CKR36 | $-kA(t) + k_q A(t)\log(\gamma t + 1)$ |

Table 4 – continued from previous page

| Domain | Equation ID | Equation |
|--------|-------------|----------|
| Biology | BPG1 | $r(1 - P(t)/K_0)P(t) + rP(t)^{0.33}$ |
| | BPG2 | $rP(t)\exp(-\gamma t) + rP(t)^2/(\alpha P(t) + 1)$ |
| | BPG3 | $\beta P(t)\sin(\omega t) + rP(t)\exp(-\gamma t)$ |
| | BPG4 | $r(-1 + P(t)/\alpha)(1 - P(t)/K_0)P(t) + r(1 - \exp(-\gamma P(t)))P(t)$ |
| | BPG5 | $r(1 - P(t)/K_0)P(t) + rP(t)/(1 + \exp(-\alpha(-\beta + P(t))))$ |
| | BPG6 | $r(1 - P(t)/K_0)P(t) + rP(t)^2/(\alpha P(t) + 1)$ |
| | BPG7 | $-Q\alpha P(t) + r(1 - P(t)/K_0)P(t) + rP(t)^{0.33} + rP(t)$ |
| | BPG8 | $r(-1 + P(t)/\alpha)(1 - P(t)/K_0)P(t) + r(1 - P(t)/K_0)P(t) + rP(t)^{0.33}$ |
| | BPG9 | $r(1 - P(t)/K_0)P(t) + rP(t)^{0.33} + rP(t)$ |
| | BPG10 | $r(-1+P(t)/\alpha)(1-P(t)/K_0)P(t)+r(1-P(t)/K_0)P(t)+r(1-\exp(-\gamma P(t)))P(t)$ |
| | BPG11 | $rP(t)^{0.33} + rP(t)$ |
| | BPG12 | $r(1 - P(t)/K_0)P(t) + rP(t)^{0.33} + rP(t)\exp(-\gamma t)$ |
| | BPG13 | $\beta P(t)\sin(\omega t) + r(1 - P(t)/K_0)P(t)$ |
| | BPG14 | $r(-1 + P(t)/\alpha)(1 - P(t)/K_0)P(t) + rP(t) + rP(t)/(1 + \exp(-\alpha(-\beta + P(t))))$ |
| | BPG15 | $r(1 - P(t)/K_0)P(t) + r(1 - \exp(-\gamma P(t)))P(t) + rP(t)\exp(-\gamma t)$ |
| | BPG16 | $rP(t)^{0.33} + rP(t)\exp(-\gamma t)$ |
| | BPG17 | $r(-1 + P(t)/\alpha)(1 - P(t)/K_0)P(t) + rP(t)^{0.33} + rP(t)$ |
| | BPG18 | $r(-1 + P(t)/\alpha)(1 - P(t)/K_0)P(t) + rP(t)^{0.33}$ |
| | BPG19 | $\beta P(t)\sin(\omega t) + r(1 - P(t)/K_0)P(t) + rP(t)$ |
| | BPG20 | $r(1 - P(t)/K_0)P(t) + rP(t)/t^{\alpha}$ |
| | BPG21 | $r(-1+P(t)/\alpha)(1-P(t)/K_0)P(t)+r(1-P(t)/K_0)P(t)+rP(t)/(1+\exp(-\alpha(-\beta+ P(t))))$ |
| | BPG22 | $r(-1 + P(t)/\alpha)(1 - P(t)/K_0)P(t) + rP(t)/t^{\alpha}$ |
| | BPG23 | $r(1 - \exp(-\gamma P(t)))P(t) + rP(t)\exp(-\gamma t)$ |
| | BPG24 | $r(1 - P(t)/K_0)P(t) + r(1 - \exp(-\gamma P(t)))P(t)$ |
| Physics | PO1 | $F_0\sin(t) - \beta\sin(v(t)) - \omega_0^2 x(t)^3 - \omega_0^2 x(t)\exp(-|x(t)|)$ |
| | PO2 | $F_0\sin(t) - \omega_0^2 x(t) - \omega_0^2 x(t)\exp(-|x(t)|)$ |
| | PO3 | $-\alpha v(t)^3 - \mu(1 - x(t)^2)v(t) - \omega_0^2 x(t) - \omega_0^2 x(t)\exp(-|x(t)|)$ |
| | PO4 | $F_0\sin(t) - \beta\sin(v(t)) - 2\beta v(t)$ |
| | PO5 | $F_0\sin(t) - \alpha v(t)^3 - \omega_0^2(\gamma|v(t)|^{0.33} + 1)x(t) - \omega_0^2 x(t)$ |
| | PO6 | $-\beta\sin(v(t)) - 2\beta v(t) - \omega_0^2(\gamma|v(t)|^{0.33} + 1)x(t) - \omega_0^2 x(t)^3 - \omega_0^2 x(t)$ |
| | PO7 | $-\beta\log(|v(t)| + 1) - 2\beta v(t) - \omega_0^2 x(t)^3$ |
| | PO8 | $-\alpha v(t)^3 - \beta|v(t)|^{0.33} - \omega_0^2 x(t)^3$ |

Table 4 – continued from previous page

| Domain | Equation ID | Equation |
|---|---|---|
| | PO9 | $-\beta|v(t)|^{0.33} - \omega_0^2 x(t)^3$ |
| | PO10 | $F_0 \sin(t) - \mu(1 - x(t)^2)v(t) - \omega_0^2(\gamma|v(t)|^{0.33} + 1)x(t) - \omega_0^2 x(t)$ |
| | PO11 | $F_0 \sin(t) - \omega_0^2(\gamma t + 1)x(t) - \omega_0^2 x(t)^3 - \omega_0^2 x(t)$ |
| | PO12 | $-\beta \sin(v(t)) - \omega_0^2(\gamma t + 1)x(t) - \omega_0^2 x(t)^3$ |
| | PO13 | $F_0 \sin(t) - \alpha v(t)^3 - \beta|v(t)|^{0.33} - \omega_0^2(\gamma t + 1)x(t) - \omega_0^2 x(t)$ |
| | PO14 | $F_0 \sin(t) - \mu(1 - x(t)^2)v(t) - \omega_0^2(\gamma|v(t)|^{0.33} + 1)x(t)$ |
| | PO15 | $F_0 \sin(t) - \beta \log(|v(t)| + 1) - \beta \sin(v(t)) - 2\beta v(t) - \mu(1 - x(t)^2)v(t)$ |
| | PO16 | $F_0 \sin(t) - \omega_0^2(\gamma|v(t)|^{0.33} + 1)x(t) - \omega_0^2 x(t) - \omega_0^2 x(t) \exp(-|x(t)|)$ |
| | PO17 | $F_0 \sin(t) - \beta \sin(x(t))v(t) - \beta \sin(v(t)) - \omega_0^2 x(t)^3$ |
| | PO18 | $F_0 \sin(t) - \beta \sin(x(t))v(t) - 2\beta v(t) - \omega_0^2 x(t)$ |
| | PO19 | $-\beta \sin(x(t))v(t) - \omega_0^2 x(t)$ |
| | PO20 | $-2\beta v(t) - \omega_0^2 x(t) \exp(-|x(t)|)$ |
| | PO21 | $-\alpha v(t)^3 - \beta \log(|v(t)| + 1) - 2\beta v(t) - \mu(1 - v(t)^2)v(t) - \omega_0^2(\gamma|v(t)|^{0.33} + 1)x(t)$ |
| | PO22 | $F_0 \sin(t) - \beta \sin(x(t))v(t)$ |
| | PO23 | $-2\beta v(t) - \beta \exp(-|x(t)|)v(t) - \mu(1 - x(t)^2)v(t) - \omega_0^2 x(t)^3$ |
| | PO24 | $F_0 \sin(t) - \beta \log(|v(t)| + 1) - \omega_0^2 x(t) \exp(-|x(t)|)$ |
| | PO25 | $F_0 \sin(t) - \alpha v(t)^3 - \beta \log(|v(t)| + 1)$ |
| | PO26 | $F_0 \sin(t) - \beta \sin(v(t))$ |
| | PO27 | $F_0 \sin(t) - \beta \log(|v(t)| + 1) - 2\beta v(t) - \omega_0^2 x(t)^3$ |
| | PO28 | $F_0 \sin(t) - \alpha v(t)^3 - 2\beta v(t) - \beta \exp(-|v(t)|)v(t)$ |
| | PO29 | $-2\beta v(t) - \omega_0^2(\gamma|v(t)|^{0.33} + 1)x(t) - \omega_0^2 x(t)^3 - \omega_0^2 x(t)$ |
| | PO30 | $-\mu(1 - x(t)^2)v(t) - \omega_0^2(\gamma t + 1)x(t) - \omega_0^2 x(t)^3$ |
| | PO31 | $-\alpha v(t)^3 - \beta \sin(x(t))v(t) - \beta \sin(v(t)) - \omega_0^2 x(t)^3$ |
| | PO32 | $-\omega_0^2(\gamma|v(t)|^{0.33} + 1)x(t) - \omega_0^2 x(t)^3$ |
| | PO33 | $F_0 \sin(t) - \alpha v(t)^3 - \beta \exp(-|v(t)|)v(t) - \omega_0^2 x(t)^3$ |
| | PO34 | $-2\beta v(t) - \mu(1 - v(t)^2)v(t) - \omega_0^2(\gamma t + 1)x(t) - \omega_0^2 x(t)$ |
| | PO35 | $-2\beta v(t) - \mu(1 - v(t)^2)v(t) - \omega_0^2(\gamma|v(t)|^{0.33} + 1)x(t)$ |
| | PO36 | $F_0 \sin(t) - \beta \sin(v(t)) - \omega_0^2(\gamma|v(t)|^{0.33} + 1)x(t)$ |
| | PO37 | $F_0 \sin(t) - \beta \exp(-|x(t)|)v(t)$ |
| | PO38 | $F_0 \sin(t) - \alpha v(t)^3 - 2\beta v(t) - \omega_0^2(\gamma t + 1)x(t)$ |
| | PO39 | $-\beta \sin(v(t)) - \mu(1 - x(t)^2)v(t) - \omega_0^2 x(t) \exp(-|x(t)|)$ |
| | PO40 | $F_0 \sin(t) - \alpha v(t)^3 - \beta \exp(-|x(t)|)v(t) - \mu(1 - v(t)^2)v(t)$ |
| | PO41 | $F_0 \sin(t) - \beta|v(t)|^{0.33} - \omega_0^2(\gamma|v(t)|^{0.33} + 1)x(t) - \omega_0^2 x(t)^3 - \omega_0^2 x(t)$ |

Table 4 – continued from previous page

| Domain | Equation ID | Equation |
|--------|-------------|----------|
| | PO42 | $-\mu(1 - x(t)^2)v(t) - \omega_0^2 x(t)\exp(-|x(t)|)$ |
| | PO43 | $F_0\sin(t) - \alpha v(t)^3 - \beta\sin(x(t))v(t) - 2\beta v(t)$ |
| | PO44 | $F_0\sin(t) - \beta\sin(x(t))v(t) - 2\beta v(t) - \mu(1 - x(t)^2)v(t) - \omega_0^2 x(t)\exp(-|x(t)|)$ |
| Material | MatSci1 | $E_0\epsilon(-\alpha_T(T - T_0) + 1) - \beta(T - T_0) + \epsilon^M\eta(T - T_0)$ |
| | MatSci2 | $H\epsilon^3 + K\epsilon^N\exp(-Q/(RT)) + \epsilon\eta\sin(T - T_0)$ |
| | MatSci3 | $H\epsilon^3 + \eta(T - T_0)\exp(-\epsilon)$ |
| | MatSci4 | $H\epsilon^3 + K\epsilon^N\exp(-Q/(RT)) + \epsilon^3\eta(T - T_0)$ |
| | MatSci5 | $E_0\epsilon^2 + \eta(T - T_0)\log(\epsilon + 1)$ |
| | MatSci6 | $E_0\epsilon(-\alpha_T(T - T_0) + 1) + K\epsilon^N\exp(-Q/(RT)) + \epsilon^M\eta(T - T_0)$ |
| | MatSci7 | $E_0\epsilon(-\alpha_T(T - T_0) + 1) + \epsilon\eta(T - T_0)^2$ |
| | MatSci8 | $H\epsilon^3 - \beta(T - T_0) + \eta(T - T_0)\log(\epsilon + 1)$ |
| | MatSci9 | $E_0\epsilon(-\alpha_T(T - T_0) + 1) + \epsilon^M\eta(T - T_0)$ |
| | MatSci10 | $H\epsilon^3 - \beta(T - T_0) + \epsilon^3\eta(T - T_0)$ |
| | MatSci11 | $H\epsilon^3 + K\epsilon^N\exp(-Q/(RT)) + \epsilon\eta(T - T_0)^2$ |
| | MatSci12 | $K\epsilon^N\exp(-Q/(RT)) + \epsilon^3\eta(T - T_0)$ |
| | MatSci13 | $E_0\epsilon(-\alpha_T(T - T_0) + 1) + K\epsilon^N\exp(-Q/(RT)) + \epsilon\eta\exp(-(T - T_0)^2)$ |
| | MatSci14 | $-\beta(T - T_0) + \epsilon\eta\exp(-(T - T_0)^2)$ |
| | MatSci15 | $-\beta(T - T_0) + \epsilon^M\eta(T - T_0)$ |
| | MatSci16 | $E_0\epsilon(-\alpha_T(T - T_0) + 1) + \epsilon\eta\exp(-(T - T_0)^2)$ |
| | MatSci17 | $E_0\epsilon^2 + \epsilon\eta(T - T_0)^2$ |
| | MatSci18 | $E_0\epsilon(-\alpha_T(T - T_0) + 1) - \beta(T - T_0) + \eta(T - T_0)\log(\epsilon + 1)$ |
| | MatSci19 | $H\epsilon^3 + \eta(T - T_0)\sin(\epsilon)$ |
| | MatSci20 | $E_0\epsilon^2 - \beta(T - T_0) + \epsilon^3\eta(T - T_0)$ |
| | MatSci21 | $E_0\epsilon^2 + \epsilon\eta\sin(T - T_0)$ |
| | MatSci22 | $K\epsilon^N\exp(-Q/(RT)) - \beta(T - T_0) + \eta(T - T_0)\log(\epsilon + 1)$ |
| | MatSci23 | $E_0\epsilon(-\alpha_T(T - T_0) + 1) + H\epsilon^3 + \eta(T - T_0)\sin(\epsilon)$ |
| | MatSci24 | $K\epsilon^N\exp(-Q/(RT)) + \epsilon\eta\sin(T - T_0)$ |
| | MatSci25 | $E_0\epsilon^2 + E_0\epsilon(-\alpha_T(T - T_0) + 1) + \eta(T - T_0)\log(\epsilon + 1)$ |

**Ground Truth:** $\dfrac{dP}{dt} = 0.9540\left(1 - \dfrac{P}{96.9069}\right)P + 0.9540P^{\frac{1}{3}}$

**a) Predicted: Direct** (Llama-3.1-8b)

```python
def equation(t: np.ndarray, P: np.ndarray, params: np.ndarray) -> np.ndarray:
    """ Mathematical function for Population growth rate

    Args:
        t: A numpy array representing observations of Time.
        P: A numpy array representing observations of Population at time t.
        params: Array of numeric constants or parameters to be optimized

    Return:
        A numpy array representing Population growth rate as the result of applying the
mathematical function to the inputs.
    """
    output = (
        params[0] * P  # A linear term based on population size
        + params[1] * P ** 2  # A squared term for birth or death ratio
        + params[2] * t + params[3] * P * t  # Linear and linear cross term
        + params[4]
    )
    return output
```

**b) Predicted: SGA** (Llama-3.1-8b)

```python
class SymbolicEquation(nn.Module):

    def __init__(self, r: float = 0.5, K: float = 100.0, time_scale: float = 0.05):
        """
        Define trainable continuous parameters for differentiable optimization.
        Tentatively initialize the parameters with the default values in args.

        Args:
            r (float): growth rate.
            K (float): carrying capacity.
            time_scale (float): time-dependent coefficient.
        """
        super().__init__()
        self.r = nn.Parameter(torch.tensor(r, dtype=torch.float))  # (1,)
        self.K = nn.Parameter(torch.tensor(K, dtype=torch.float))  # (1,)
        self.time_scale = nn.Parameter(torch.tensor(time_scale, dtype=torch.float))

        # # (1,)
        # self.r = nn.Parameter(torch.tensor(r, dtype=torch.float))

    def forward(self, t: torch.Tensor, P: torch.Tensor) -> torch.Tensor:
        """"Mathematical function for Population growth rate

        Args:
            t: Time.
            P: Population at time t.

        Return:
            dP_dt: Population growth rate.
        """
        # # (B,)
        dP_dt = self.r * torch.exp(-self.time_scale*t) * P * (1 - P / self.K)  # (B,)
        return dP_dt
```

**c) Predicted: LaSR** (Llama-3.1-8b)

$$\frac{dP}{dt} = \left(-910.31 - \left(\left(P \cdot (P + \sqrt{P})\right) + \log\left((t + 4.6396) - (-0.66095)\right)\right) \cdot \sin\left(\frac{-0.95572}{e^t - t}\right)\right) + 910.3$$

**d) Predicted: LLM-SR** (Llama-3.1-8b)

```python
def equation(t: np.ndarray, P: np.ndarray, params: np.ndarray) -> np.ndarray:
    """
    Mathematical function for Population growth rate with a logistic growth model, decay,
    and periodicity.

    Args:
        t: A numpy array representing observations of Time.
        P: A numpy array representing observations of Population at time t.
        params: Array of numeric constants or parameters to be optimized (k, M, r, a, b, A,
w, tau)

    Returns:
        A numpy array representing Population growth rate as the result of applying the
mathematical function to the inputs.
    """
    # Apply transformations to inputs (if needed)
    t = t / np.max(t)  # Normalize time
    P = P / np.max(P)  # Normalize population

    # Compute the rates using logistic growth model
    dP_dt = params[0] * P * (1 - P) - params[1] * P - params[2] * P**params[3]

    # Compute the periodicity factor
    periodic_factor = params[4] * np.sin(params[5] * (t - params[6]) + params[7])

    # Combine growth rate and periodicity
    dP_dt_with_periodicity = dP_dt * periodic_factor

    return dP_dt_with_periodicity
```

*Figure 14.* Example of output hypotheses from different LLM scientific equation discovery methods for BPG0 problem in LSR-Synth biology domain.

**Ground Truth:**

$$0.19 \cdot A(t) + 0.19 \cdot \cos\left(\log(A(t) + 1)\right)$$

**a) Predicted: Direct** (Llama-3.1-8b)

```python
def equation(t: np.ndarray, A: np.ndarray, params: np.ndarray) -> np.ndarray:
    """ Mathematical function for Rate of change of concentration (Growth, Reaction or
Decay)

    Args:
        t: A numpy array representing observations of Time.
        A: A numpy array representing observations of Concentration at time t.
        params: Array of numeric constants or parameters to be optimized.
                - params[0]: k (Rate constant, e.g., 0.1 min^-1)
                - params[1]: C0 (initial concentration, e.g., 10) for decay or growth
models

    Return:
        A numpy array representing Rate of change of concentration as the result of
applying the mathematical function to the inputs.
        - equation implemented under assumption that k*A represents linear decay, or
linear growth.
    """
    output = params[0] * A + params[1] * params[2]
    return output
```

**b) Predicted: SGA** (Llama-3.1-8b)

```python
class SymbolicEquation(nn.Module):

    def __init__(self, k: float = 9.00e-02, A_0: float = 9.60e-01, A_t: float =
9.60e-01):
        """
        Define trainable continuous parameters for differentiable optimization.
        Tentatively initialize the parameters with the default values in args.

        Args:
            k (float): time-dependent reaction rate constant.
            A_0 (float): initial concentration.
            A_t (float): time-dependent change in concentration.
        """
        super().__init__()
        # (1,)
        self.k = nn.Parameter(torch.tensor(k, dtype=torch.float))
        self.A_0 = nn.Parameter(torch.tensor(A_0, dtype=torch.float))   # (1,)
        self.A_t = nn.Parameter(torch.tensor(A_t, dtype=torch.float))   # (1,)

    def forward(self, t: torch.Tensor, A: torch.Tensor) -> torch.Tensor:
        """Mathematical function for Rate of change of concentration in chemistry
reaction kinetics

        Args:
            t: Time.   # (B, 1)
            A: Concentration at time t.   # (B, 1)

        Return:
            dA_dt: Rate of change of concentration in chemistry reaction kinetics.   #
(B, 1)
        """
        # (B, 1)
        dA_dt = self.k * A + self.A_0 * self.A_t * t   # (B, 1)
        return dA_dt
```

**c) Predicted: LaSR** (Llama-3.1-8b)

$$\left(-0.0040402/(1.3222^t)\right) * \left((A^t)^{cos\left(\left(\left(\left(t+((A/(A/A))-0.043477))^A\right)-A\right)+((0.75629^{1.2028})^t)\right)\right)}\right)$$

**d) Predicted: LLM-SR** (Llama-3.1-8b)

```python
def equation(t: np.ndarray, A: np.ndarray, params: np.ndarray) -> np.ndarray:
    """ Mathematical function for Rate of change of concentration in chemistry reaction
kinetics

    Args:
        t: A numpy array representing observations of Time.
        A: A numpy array representing observations of Concentration at time t.
        params: Array of numeric constants or parameters to be optimized

    Return:
        A numpy array representing Rate of change of concentration in chemistry reaction
kinetics as the result of applying the mathematical function to the inputs.
    """
    output = params[0] * A + params[1] * A**2 + params[2] * A**3 + params[3] * t +
params[4]
    return output
```

*Figure 15.* Example of output hypotheses from different LLM scientific equation discovery methods for CKR2 problem in LSR-Synth chemistry domain.

**Ground Truth:**

$$31.12 \cdot \varepsilon \cdot (-0.05 \cdot (T - 316.19) + 1) - 0.23 \cdot (T - 316.19) + \varepsilon^{0.60} \cdot 3.63 \cdot (T - 316.19)$$

**a) Predicted: Direct** (Llama-3.1-8b)

```python
def equation(epsilon: np.ndarray, T: np.ndarray, params: np.ndarray) -> np.ndarray:
    """ Mathematical function for Stress

    Args:
        epsilon: A numpy array representing observations of Strain.
        T: A numpy array representing observations of Temperature.
        params: Array of numeric constants or parameters to be optimized

    Return:
        A numpy array representing Stress as the result of applying the mathematical
function to the inputs.
    """
    reference_temperature = 0
    sigma_0 = params[0]
    alpha = params[1]
    n = params[2]
    linear_epsilon_term = params[3] * epsilon
    multiplicative_epsilon_term = params[4] * epsilon
    linear_temperature_term = params[5] * T
    output = sigma_0 * (1 + alpha * (T - reference_temperature)) * epsilon**n +
linear_epsilon_term + multiplicative_epsilon_term + linear_temperature_term
    return output
```

**b) Predicted: SGA** (Llama-3.1-8b)

```python
class SymbolicEquation(nn.Module):

    def __init__(self, c_0: float = 9.83e-01, c_1: float = 9.86e-01, c_2: float =
9.83e-01,
                 a: float = -3.73e-03, b: float = -1.11e-02, d: float = 2.63e-02):
        """
        Define trainable continuous parameters for differentiable optimization.
        Tentatively initialize the parameters with the default values in args.

        Args:
            c_0 (float): coefficient.
            c_1 (float): coefficient.
            c_2 (float): coefficient.
            a (float): coefficient for quadratic term.
            b (float): coefficient for exponential term.
            d (float): coefficient for non-linear relationship between Strain and
Temperature.
        """
        super().__init__()
        self.c_0 = nn.Parameter(torch.tensor(c_0, dtype=torch.float))  # (1,)
        self.c_1 = nn.Parameter(torch.tensor(c_1, dtype=torch.float))  # (1,)
        self.c_2 = nn.Parameter(torch.tensor(c_2, dtype=torch.float))  # (1,)
        self.a = nn.Parameter(torch.tensor(a, dtype=torch.float))  # (1,)
        self.b = nn.Parameter(torch.tensor(b, dtype=torch.float))  # (1,)
        self.d = nn.Parameter(torch.tensor(d, dtype=torch.float))  # (1,)

    def forward(self, epsilon: torch.Tensor, T: torch.Tensor) -> torch.Tensor:
        """Mathematical function for Stress

        Args:
            epsilon: Strain. # (B, 1)
            T: Temperature. # (B, 1)

        Return:
            sigma: Stress. # (B, 1)
        """
        # (B, 1)
        sigma = self.c_0 + self.c_1 * torch.exp(self.a * epsilon + self.b * T) +
self.c_2 * torch.cos(self.d * epsilon * T)
        return sigma
```

**c) Predicted: LaSR** (Llama-3.1-8b)

$$\left( \left( \sqrt{\epsilon} \cdot (-712.36) + \left( \left( \frac{\sqrt{\epsilon}}{1.3792^{\epsilon}} \cdot 2.2798 \right) \cdot (T - \epsilon) + \epsilon \right) + 6.8125 \right) \cdot 1.5076^{\sqrt{\epsilon}} \right) - \sin(\log(T))$$

**d) Predicted: LLM-SR** (Llama-3.1-8b)

```python
def equation(epsilon: np.ndarray, T: np.ndarray, params: np.ndarray) -> np.ndarray:
    """ Mathematical function for Stress

    Args:
        epsilon: A numpy array representing observations of Strain.
        T: A numpy array representing observations of Temperature.
        params: Array of numeric constants or parameters to be optimized

    Return:
        A numpy array representing Stress as the result of applying the mathematical
function to the inputs.
    """
    output = params[0] * (epsilon ** (1/params[4]) - 1) + \
             params[1] * (epsilon ** (1/params[6]) - 1) + \
             params[2] * (epsilon ** (1/params[8]) - 1) * np.exp(params[3] * epsilon) + \
             params[5] * epsilon + params[6] * T + params[7]
    return output
```

*Figure 16.* Example of output hypotheses from different LLM scientific equation discovery methods for MatSci0 problem in LSR-Synth material science domain.

**Ground Truth:**

$$F_0 \cdot \sin(t) - \beta \cdot \sin(v(t)) - \omega_0^2 \cdot x(t)^3 - \omega_0^2 \cdot x(t) \cdot e^{-|x(t)|}$$

**a) Predicted: Direct** (Llama-3.1-8b)

```python
def equation(x: np.ndarray, t: np.ndarray, v: np.ndarray, params: np.ndarray) ->
np.ndarray:
    """ Mathematical function for Acceleration

    Args:
        x: A numpy array representing observations of Position at time t.
        t: A numpy array representing observations of Time.
        v: A numpy array representing observations of Velocity at time t.
        params: Array of numeric constants or parameters to be optimized

    Return:
        A numpy array representing Acceleration as the result of applying the
mathematical function to the inputs.
    """
    # Since acceleration is Force/mass and Force is mass*velocity change,
    # we represent this equation as
    # a = dv/dt or in our representation format, given our position x it will correspond
to
    # a = d^2x/dt^2 or by simple numeric analogy 4th order approximation involving high
degree dependency functions:
    output = (params[0]*x**4 + params[1]*x**3 + params[2]*x**2 + params[3]*x +
params[4]) + params[5]*v**2 + params[6]*t + params[7]*(x**2 + v)
    return output
```

**b) Predicted: SGA** (Llama-3.1-8b)

```python
class SymbolicEquation(nn.Module):

    def __init__(self, c_0: float = 9.50e-01, c_1: float = 9.50e-01, c_2: float =
9.50e-01, c_3: float = 1.05e+00):
        """
        Define trainable continuous parameters for differentiable optimization.
        Tentatively initialize the parameters with the default values in args.

        Args:
            c_0 (float): coefficient.
            c_1 (float): coefficient.
            c_2 (float): coefficient.
            c_3 (float): coefficient.
        """
        super().__init__()
        self.c_0 = nn.Parameter(torch.tensor(c_0, dtype=torch.float))  # (1,)
        self.c_1 = nn.Parameter(torch.tensor(c_1, dtype=torch.float))  # (1,)
        self.c_2 = nn.Parameter(torch.tensor(c_2, dtype=torch.float))  # (1,)
        self.c_3 = nn.Parameter(torch.tensor(c_3, dtype=torch.float))  # (1,)

    def forward(self, x: torch.Tensor, t: torch.Tensor, v: torch.Tensor) ->
torch.Tensor:
        """Mathematical function for Acceleration in Nonl-linear Harmonic Oscillator

        Args:
            x: Position at time t. # (B, )
            t: Time. # (B, )
            v: Velocity at time t. # (B, )

        Return:
            dv_dt: Acceleration in Nonl-linear Harmonic Oscillator. # (B, )
        """
        # Non-linear relationship between x, t, and v
        dv_dt = self.c_0 * torch.exp(-self.c_1 * x) + self.c_2 * torch.cos(self.c_3 *
t + self.c_3 * x)    # (B, )
        return dv_dt
```

**c) Predicted: LaSR** (Llama-3.1-8b)

$$\left(\frac{\left((x+x)\cdot\left(\left(\frac{\sqrt{t^{1.9034}-1.4311}}{x}\right)\cdot\sin(1.1478\cdot t)-x\right)\cdot\sin(x)\right)}{1.7052}\right)-\sin(0.0032827)$$

**d) Predicted: LLM-SR** (Llama-3.1-8b)

```python
def equation(x: np.ndarray, t: np.ndarray, v: np.ndarray, params: np.ndarray) ->
np.ndarray:
    """ Mathematical function for Acceleration

    Args:
        x: A numpy array representing observations of Position at time t.
        t: A numpy array representing observations of Time.
        v: A numpy array representing observations of Velocity at time t.
        params: Array of numeric constants or parameters to be optimized

    Return:
        A numpy array representing Acceleration as the result of applying the mathematical
function to the inputs.
    """
    # Since acceleration is Force/mass and Force is mass*velocity change,
    # we represent this equation as
    # a = dv/dt or in our representation format, given our position x it will correspond
to
    # a = d^2x/dt^2 or by simple numeric analogy 4th order approximation involving high
degree dependency functions:
    output = (params[0]*x**4 + params[1]*x**3 + params[2]*x**2 + params[3]*x + params[4])
+ params[5]*v**2 + params[6]*t + params[7]*(x**2 + v)
    return output
```

*Figure 17.* Example of output hypotheses from different LLM scientific equation discovery methods for PO0 problem in LSR-Synth physics domain.

