# OpenReview forum: "LLM-SRBench: A New Benchmark for Scientific Equation Discovery with Large Language Models"
_ICML.cc/2025/Conference — ICML 2025 oral_

### Official Review · Reviewer_tSk5 · 2025-03-12

**Overall Recommendation:** 3

**Summary:**

The paper introduces a benchmark for scientific equation discovery, where the model are able to use both input/output values along with a problem description in human language, into constructing an equation that describes the data well. The model tested on this benchmark would be measured by the accuracy of the recovered equation and also the semantic accuracy of the equation. From results, normal LLMs are able to perform poorly on this task.

**Claims And Evidence:**

I think based on the paper, it can be shown that LLMs are still unable to properly incorporate scientific information into its predictions, which I think is a fair conclusion from the trial experiments ran with the benchmarks.

**Essential References Not Discussed:**

None that I am aware of.

**Experimental Designs Or Analyses:**

The results seem okay, but would also benefit from some kind of confidence/score variance value as well. Some issues are with evaluation criteria, which I describe above.

**Methods And Evaluation Criteria:**

I think the dataset is quite unique, and it is definitely a good start to other works in this area. Some general comments I have with the proposed dataset are as follows:

- A point that would make the dataset more realistic and fitting of real world scenarios would be incorporation of some observation noise, which is typical when actual experiments are conducted. This could potentially be done through curation of some real experiment data, or can simply be addition of some noise into the training set.

- I notice that the preliminary results are mostly for LLM evaluators, which make sense since the benchmark also include domain knowledge in human language. Despite this, it would be interesting to see how traditional symbolic regression methods would fare on the dataset, even if they are not able to incorporate the domain knowledge. Based on what the metrics from the authors, they may fare well in terms of accuracy but less so in semantic similarity, which would make the need for LLM-based discovery stronger, and the dataset more useful.

Also some comments regarding evaluation criteria:

- I'm not fully convinced with using LLMs to evaluate semantic similarity, also despite the 94.6% figure quoted by the authors whose experimental details are not so clear (e.g., how can we be sure the 130 test cases cover all cases that can be output by an LLM?). I understand that it can be hard to determine semantic similarity, but maybe more classical methods would be applicable if the output form of the equations can be more restrictive (which may be against what the authors want though).

- From what I understand, typical SR papers uses R2 score as the evaluation criteria. I'm wondring why this paper does not also include that.

**Other Comments Or Suggestions:**

None.

**Other Strengths And Weaknesses:**

None.

**Questions For Authors:**

Most of the questions/suggestions I have additionally can be found in previous sections, and are related to some aspects of the evaluation criteria, and with the realism of the dataset that may be improved.

**Relation To Broader Scientific Literature:**

I think the paper would have great relevance, since it can be a good step to involve scientific reasoning into modelling data. Even though there are still ways that the benchmark could be improved to be more realistic with the scientific uses, or to incorporate even more equations from more domains within each subfields, but overall it may still be a good start for work in this direction.

**Theoretical Claims:**

There are none.

---

> ### Author Rebuttal · Authors · 2025-04-01
>
> Thank you for dedicating your time and expertise to review our submission. Please find our responses below.
> > it would be interesting to see how traditional symbolic regression methods would fare on the dataset... which would make the need for LLM-based discovery stronger, and the dataset more useful.
>
> Thanks for the thoughtful suggestion and constructive feedback. Previous work on LLM-based scientific equation discovery (LLM-SR, LaSR, and SGA) has provided evidence for the benefits of these approaches compared to traditional methods. However, we agree that evaluating traditional non-LLM methods on our benchmark would further demonstrate these advantages.
> In response to your suggestion, we conducted experiments with PySR (a state-of-the-art non-LLM baseline) on our benchmark datasets during the rebuttal period. The results (shown in table below) demonstrate that PySR often achieves competitive numeric accuracy but considerably lower symbolic accuracy, particularly on non-physics datasets. This confirms that while traditional methods can fit data numerically, they struggle with symbolic understanding due to lack of domain knowledge. We also think that these findings further motivate the value of LLM-based techniques for scientific equation discovery. Thank you again for the constructive feedback. We will surely include this analysis in the final version.
>
> | Dataset (Metric) | LLM-SR (best) | LaSR (best) | SGA (best) | PySR |
> |---------|---------------|-------------|------------|------|
> | LSR-Transform (SA / Acc0.1) | 31.53 / 39.64 | 12.61 / 50.45 | 9.91 / 8.11 | 8.11 / 56.76 |
> | LSR-Synth Chemistry (SA / Acc0.1) | 11.11 / 66.66 | 2.77 / 38.92 | 0 / 16.66 | 0 / 41.67 |
> | LSR-Synth Biology (SA / Acc0.1) | 25.30 / 58.33 | 8.33 / 20.83 | 4.16 / 12.51 | 0 / 25.0 |
> | LSR-Synth Physics (SA / Acc0.1) | 9.91 / 36.36 | 9.91 / 31.81 | 4.54 / 9.09 | 4.54 / 29.55 |
> | LSR-Synth MatSci (SA / Acc0.1) | 20.24 / 88.28 | 28.12 / 72.04 | 0 / 36.11 | 0 / 68.0 |
>
> **Incoporation of noise** Thank you for the comment. We have not explored the imacpt of noise in the our benchmark's data generation. The main motivation of this benchmark was to help community towards building better general LLM-based equation discovery agents that can be leveraged in scientific domains, addressing the challenges of current benchmarks for the emerging LLM-based techniques.
> We fully agree with the reviewer on the importance of noise in real-world scientific discovery scenarios. This is one of the directions we're considering for future benchmark enhancements.
>
> **R2 vs Accuracy to Tolerance** Thanks for raising this important question. Based on our analysis (and some of the previous works [Kamienny et al., 2022; Biggio et al., 2021]), R2 is not a good metric for the evaluation of symbolic regression, particularly if we have large-scale synthetic data with very different output scales.
> - R2 can be easily saturated by normalized mean patterns (similar to NMSE), often missing prediction nuances at different scales. This is evidenced by the consistently high R2 scores (above 0.999) achieved by most recent methods in benchmarks like SRBench.
> - The accuracy-to-tolerance metric (defined in Section 2.3) aggregates point-wise normalized metrics for each datapoint rather than normalizing mean aggregated scores. It also implements a tolerance threshold for the worst point-wise normalized distance, making evaluation more robust to function behavior nuances (e.g., sudden changes or spikes)
> As the goal of symbolic regression is to learn correct underlying mathematical relations, we should care more about these nuances of function numeric behavior as well as the symbolic mathematical alignment which makes accuracy to tolerance a better metric for numeric precision assessment.
>
> **Evaluation of Equation Semantic Similarity with LLMs vs more classical methods** Thank you for the comment. We will surely add more details on equation semantic similarity evaluation in the updated version.
> Regarding the question, we think that restricting the outputs of the equations to specific forms for LLM-based equation discovery won’t be a good idea since one of the main benefits of LLMs is their capabilities in programming and code generation which opens new more flexible representations for equation discovery that was not possible with the previous expression tree based methods and their corresponding symbolic evaluations.
> While evaluating semantic similarity is non-trivial, our empirical analysis and human study show that LLMs are usually strong at recognizing equation equivalence across different styles and representations. We also think that correlation between symbolic accuracy (computed from GPT-4o) and generalization (computed from OOD data) shown in Figure 6 further support LLMs effectiveness as semantic symbolic evaluators for equation discovery.
>
> ---
> We hope that most of the reviewer’s concerns have been addressed. We’d be happy to engage in further discussions.

---

### Official Review · Reviewer_vLgS · 2025-03-13

**Overall Recommendation:** 3

**Summary:**

The paper introduces LLM-SRBench, a novel benchmark designed to evaluate the capabilities of LLMs in scientific equation discovery. The key motivation behind the benchmark is to prevent trivial memorization by LLMs, which has been a limitation in existing equation discovery benchmarks. The benchmark consists of 239 equation discovery tasks across 4 scientific domains (physics, chemistry, biology, and material science) and is structured into two different categories:

LSR-Transform: Reformulates well-known physical equations, from the Feynman benchmark, into less common mathematical forms, making it more difficult for LLMs to rely on memorization.

LSR-Synth: Introduces synthetic equations that combine known scientific terms with novel, discovery-driven components, requiring reasoning and data-driven inference.

The paper evaluates multiple SOTA LLM-based equation discovery methods, including LLM-SR, LaSR, SGA, and direct prompting baselines, using both open-source (Llama-3.1-8B) and closed-source (GPT-4o-mini, GPT-3.5) models. To evaluate model performance, the paper also proposes a novel symbolic accuracy metric, using GPT-4o as an evaluator. Symbolic accuracy correlates with out-of-domain (OOD) performance, suggesting that correctly identified symbolic structures contribute to better generalization. The best-performing method achieves only 31.5% symbolic accuracy on LSR-Transform, highlighting the difficulty of the task. Direct prompting (without data-driven reasoning) performs poorly, showing the necessity of data feedback in scientific equation discovery. LLMs struggle with OOD generalization, indicating that discovered equations often fail to extrapolate beyond the training data.

**Claims And Evidence:**

Supported claims:

The following claims are well-supported by empirical evidence provided in the paper:

1) The authors demonstrate that existing equation discovery benchmarks allow LLMs to rely on memorization rather than true discovery by showing significant performance drops when problems are reformulated (Figure 1).
2) The claim that LLM-SRBench is more challenging is well-supported by the low symbolic accuracy scores (Table 1).
3) The evidence for poor OOD generalization is strong, as models show significantly higher NMSE on OOD test sets (Figure 5), reinforcing the difficulty of extrapolation.


Claims That Need Stronger Justification:

The paper provides evidence that generalization performance varies across scientific domains, with chemistry and biology showing a larger performance gap between in-domain (ID) and OOD cases compared to physics and material science. However, the authors do not analyze why this variation occurs. While it is possible that some domains inherently pose more complex challenges for equation discovery, another explanation could be that the dataset design itself contributes to the difficulty, either through the structure of equations, the nature of data distributions, or the frequency of certain mathematical patterns.

A systematic classification of failure cases across different domains would provide valuable insights. Examining where and how models fail—whether due to misidentified variables, incorrect functional forms, or numerical instability—could help determine whether LLMs struggle more in certain disciplines due to intrinsic domain complexity or due to dataset-specific biases. Table 1 and Figure 5 already indicate that different models perform differently across domains, but the paper does not include concrete examples of incorrect outputs or categorize failure modes. A deeper analysis, including sample failure cases and their classification by domain and model type, could reveal whether generalization failures follow a consistent pattern (e.g., do chemistry problems frequently lead to overfitting, while physics problems lead to missing terms?).

To strengthen this point, the authors should consider adding:

- Representative examples of failure cases across domains (e.g., incorrect equations generated by benchmark models).
- A classification of common failure types (e.g., missing terms, incorrect exponents, incorrect dependencies).
- A discussion on whether these errors stem from domain-specific challenges or dataset characteristics (e.g., certain equation structures in some domains being more prone to errors).

**Essential References Not Discussed:**

No, the paper appropriately cites and discusses the relevant prior work in symbolic regression, LLM-based scientific discovery, and existing benchmarks to the best of my knowledge.

**Experimental Designs Or Analyses:**

The experimental design is sound, with clear methodology and appropriate evaluation metrics. The hyperparameters used for different methods (LLM-SR, LaSR, SGA, and Direct Prompting) are detailed in the appendix, ensuring reproducibility. Each step in the LSR-Transform and LSR-Synth pipelines is well-documented, and the authors provide code detailing the dataset generation and evaluation procedures, further supporting transparency. No major issues were found in the experimental design.

**Methods And Evaluation Criteria:**

The paper benchmarks state-of-the-art LLM-based scientific equation discovery methods using three LLM backbones (Llama-3.1-8B, GPT-3.5, and GPT-4o) and evaluates their performance on LLM-SRBench. The evaluation criteria, including novel symbolic accuracy, numerical precision, NMSE, and OOD generalization, are well-aligned with the problem of equation discovery. Additionally, the symbolic accuracy evaluation using GPT-4o is verified against human expert judgments, ensuring reliability in assessing equation correctness.

By comparing LLM-SRBench with existing benchmarks, the authors effectively demonstrate its increased difficulty and reduced sensitivity to memorization. The models evaluated include LLM-SR, LaSR, SGA, and Direct Prompting (DataBlind), providing a comprehensive comparison of different equation discovery approaches.

**Other Comments Or Suggestions:**

None.

**Other Strengths And Weaknesses:**

The strengths and weaknesses of the paper are thoroughly discussed in the "Claims And Evidence," "Methods And Evaluation Criteria," and "Questions For Authors" sections above. These include the paper’s well-structured benchmark, strong experimental design, and clear evaluation criteria, along with areas needing further justification, such as failure case analysis, novelty verification.

The LSR-Transform dataset is built upon the Feynman benchmark dataset, but the authors introduce transformations and natural language specifications for each reformulated problem, making it more challenging and reducing reliance on memorization. The LSR-Synth dataset aims to incorporate novel synthetic terms into existing equations, creating discovery-driven problems that go beyond simple equation recall. This approach has the potential to be extended beyond the 4 scientific domains used in the paper, making it a valuable resource for broader scientific applications.

**Questions For Authors:**

•  Domain-Specific Generalization Performance
The paper provides evidence that generalization performance varies across scientific domains, with chemistry and biology showing a larger performance gap between in-domain (ID) and out-of-domain (OOD) cases compared to physics and material science. However, it does not analyze why this variation occurs. Could the observed difficulty stem from intrinsic domain complexity, or might it be influenced by dataset-specific factors such as equation structure, data distribution, or frequency of certain mathematical patterns? Systematic classification of failure cases across different domains could help clarify whether these challenges are due to intrinsic scientific difficulty or dataset artifacts? Specifically:
1) Can the authors provide representative examples of failure cases across domains?
2) Would a classification of common failure types (e.g., missing terms, incorrect exponents, incorrect dependencies) help reveal domain-specific trends?
3) Could the authors discuss whether the failure modes observed in different disciplines stem from the nature of scientific equations themselves or from dataset characteristics?
Clarifying this aspect would help determine whether performance disparities across domains are inherent or dataset-driven. Please refer to the "Claims That Need Stronger Justification" section for detailed concerns.
•  Novelty Check in LSR-Synth Pipeline
In the LSR-Synth pipeline, the authors use GPT-4o as a novelty evaluator to determine whether a generated equation is distinct from known scientific expressions. However, LLMs may incorrectly classify an equation as novel despite it existing in prior literature. Since model performance on this dataset is poor, it suggests the problems are challenging, but this does not guarantee that the novelty check is reliable.
4) Could the authors justify why asking GPT-4o alone is sufficient for novelty evaluation? Would a secondary human expert review or detailed literature analysis improve the reliability of the novelty check?
5) Given that novelty assessment is crucial for ensuring LSR-Synth problems do not introduce trivial cases, how can the authors ensure that LLM-generated novelty assessments are not biased or incorrect?
•  Dataset Size Reduction in LSR-Transform
In Section 2.1 LSR-Transform, the authors state:
"This process yields 111 total transformed equations derived from the 100 original Feynman problems."
The dataset generation pipeline includes steps to increase diversity by selecting a new target variable (Step 2: Select Pivot Variable) and switching the roles of input-output variables (Step 3: Feature-Target Transformation). However, despite this expansion, only 111 transformed equations remain from an original set of 100 Feynman equations.
6) Is this reduction due to eliminations in Step 5 (Solvability Check) or Step 6 (Dataset Refinement)? Could the authors provide a breakdown of the proportion of equations discarded at each stage to better understand why the final dataset size remains close to the original? Understanding this filtering process would clarify how much of the dataset reduction is due to analytical intractability versus imposed constraints on the transformed dataset.

**Relation To Broader Scientific Literature:**

The paper builds on prior work in symbolic regression (e.g., AI Feynman, PySR) and LLM-based scientific discovery, addressing the issue of memorization in existing benchmarks. It extends benchmarks like SRBench and SRSD by introducing LSR-Transform and LSR-Synth, which focus on reasoning beyond recall. The study aligns with recent advances in LLM-guided symbolic regression (e.g., LLM-SR, LaSR, SGA) and contributes a evaluation framework to test equation discovery in a more challenging and diverse setting. By systematically testing LLMs on equation discovery tasks requiring reasoning over data rather than recall, this work helps advance our understanding of LLMs' ability to generalize mathematical structures, a crucial step toward automated scientific discovery, as demonstrated across four domains: physics, chemistry, biology, and material science.

**Theoretical Claims:**

No theoretical justification is required in this paper, as it primarily focuses on empirical benchmarking rather than formal proofs. The evaluation metrics used, including symbolic accuracy, NMSE, and OOD generalization, are well-defined and appropriate for assessing scientific equation discovery. No issues were found with their formulation or application.

---

> ### Author Rebuttal · Authors · 2025-04-01
>
> Thank you for dedicating your time and expertise to review our submission. Please find our responses below.
>
> > A systematic classification of failure cases across different domains would provide valuable insights. Examining where and how models fail—whether due to misidentified variables, incorrect functional forms, or numerical instability
>
> Thank you for the constructive suggestion. We examined the failure cases during rebutal period and noticed that LLM-based models usually don't fail due to the syntax function errors (such as missing variables or dependencies). Instead, most errors come from incorrect terms or combination of terms within the equations, reflecting mistakes due to semantic aspects of discovery for a given scientific problem. We could not identify a uniform pattern of specific mathematical relations that consistently challenge current discovery methods. We believe this is because LLM-based equation discovery involves a complex interplay of domain knowledge, search capabilities with data-driven feedback, and mathematical manipulation skills. To thoroughly study failure case patterns would require investigating all these dimensions. We agree with the reviewer on the importance of this research direction for future work.
>
>
> **Representative Failure Examples** Thank you for this suggestion. We will include examples of failure cases for different methods across domains in the camera-ready version.
>
>
> > The paper provides evidence that generalization performance varies across scientific domains, ... However, it does not analyze why this variation occurs.
>
> Thanks for the thoughtful question. We have explored problem complexity across domains (Figure 10 in Appendix shows the distribution by complexity). While biology and physics problems have similar and slightly higher complexity than other domains, we observe a smaller ID-OOD gap for physics but a larger one for biology in the results. This suggests factors beyond mere complexity are at play (like LLM domain knowledge).
> The reviewer raises an interesting point that would require domain-specific analysis beyond our study's scope. Our primary goal is to develop a benchmark that helps the community build better general LLM-based equation discovery agents applicable across domains. More specialized applications would require domain-focused analyses
>
>
> > Novelty Check...In the LSR-Synth pipeline, the authors use GPT-4o as a novelty evaluator to determine whether a generated equation is distinct from known scientific expressions. However, LLMs may incorrectly classify an equation as novel despite it existing in prior literature. .. this does not guarantee that the novelty check is reliable.
>
> We cannot theoretically guarantee whether GPT-4o evaluation of novelty is reliable, but evaluating novelty is a non trivial task and we think that LLMs (with their vast knowledge about literature) could be a helpful tool for novelty assessment. Our empirical evidence also supports the effectiveness of this approach: the consistently lower performance of discovery methods on LSR-Synth datasets suggests these problems aren't trivial and do contain novel elements (if they were simply known equations, LLM-based discovery models would likely recall them from embedded knowledge and solve them easily). We agree that more rigorous novelty evaluation involving domain experts and specialized scientific literature retrieval tools would enhance the design of benchmark problems. However, this extension would require significant domain knowledge and human expert review which is not feasible for the rebuttal period but valuable for future work.
>
>
> **Dataset Size Reduction in LSR-Transform** Here is the detailed breakdown of our filtering process in LSR-Transform: After transformation steps in Figure 3 (Step 4), we obtained 471 transformed problems from the 100 original Feynman problems; During the solvability check with sympy (Step 5), 53 problems were discarded, leaving 418 problems; In dataset refinement (Step 6), we only filtered datapoints to ensure they were within the domain of the new transformed equations, without eliminating any equations at this stage. We then filtered out 307 problems due to significantly higher complexity compared to original problems, resulting in the final 111 problems in LSR-Transform. This is to ensure that the challenging nature of LSR-Transform stems from semantic aspects of discovery rather than from syntactically more complex or lengthy problems (as shown in Figures 4 and 8). We will clarify this process further in the revised version of paper.
>
>
> **Additional Qualitative Examples of Outputs** As some reviewers have higlighted examples of Figure 14 helpful, we plan to include more hypothesis examples from different domains in the revision of paper.
>
> ---
> We hope that most of the reviewer’s concerns have been addressed. We’d be happy to engage in further discussions.

---

> > ### Comment · Reviewer_vLgS · 2025-04-05
> >
> > The authors have addressed most of my concerns appropriately. Their explanation of the dataset size reduction in the LSR-Transform pipeline was helpful, clarifying the multi-stage filtering process and their rationale behind discarding high-complexity equations. The inclusion of representative failure cases and qualitative outputs in the camera-ready version also responds to my suggestions for deeper insight into model behavior. The authors mentioned that they plan to include more hypothesis examples from different domains and will provide examples of failure cases for different methods across domains in the final version, which further strengthens their response.
> >
> > However, the use of GPT-4o as the sole novelty evaluator in the LSR-Synth pipeline remains a concern. While the authors argue that consistently poor performance on LSR-Synth suggests the problems are indeed novel, relying solely on an LLM without external verification is still problematic, especially given ongoing research questioning LLMs’ ability to judge novelty or creativity.

---

> > > ### Author Response · Authors · 2025-04-05
> > >
> > > Dear reviewer,
> > > We appreciate your thoughtful review, and we are glad our rebuttal addressed most of your concerns. You raise a valid point about using GPT-4o as the sole novelty evaluator - this is indeed a limitation we acknowledge. Novelty assessment is genuinely challenging, and while LLM performance patterns suggest them to be helpful, independent verification would strengthen our claims. This is definitely a valuable direction for future work that would benefit from domain expert involvement. Thank you again for your constructive feedback throughout this process.

---

### Official Review · Reviewer_Yany · 2025-03-14

**Overall Recommendation:** 4

**Summary:**

This paper introduces LLM-SRBench, a benchmark designed to evaluate LLMs on scientific equation discovery tasks. The authors identify a key problem: existing benchmarks like Feynman equations can be solved by LLMs through memorization rather than actual discovery. To address this, they develop two benchmark categories: (1) LSR-Transform, which transforms common equations into less familiar mathematical forms, and (2) LSR-Synth, which creates novel synthetic equations that combine known scientific terms with plausible synthetic components. The benchmark spans 239 challenging problems across chemistry, biology, physics, and material science domains. They show that state-of-the-art methods achieve only 31.5% symbolic accuracy, highlighting significant challenges in this field.

**Claims And Evidence:**

The paper's main claims are well-supported by the evidence presented. The authors convincingly demonstrate the memorization issue through Figure 1, showing how performance on standard Feynman problems exhibits patterns consistent with memorization (sharp error drops and lower symbolic error rates) rather than actual reasoning and discovery. The experiments across multiple methods and LLM backbones provide strong evidence that their benchmark is substantially more challenging than existing ones. The performance analysis across different scientific domains, complexity levels, and generalization capabilities is thorough and supports their conclusions.

**Essential References Not Discussed:**

None that I'm aware of.

**Experimental Designs Or Analyses:**

The experimental design is comprehensive and well-executed. The authors evaluate four different methods (Direct Prompting, SGA, LaSR, LLM-SR) using three different LLM backbones (Llama-3.1-8B-Instruct, GPT-4o-mini, GPT-3.5-turbo) with standardized conditions (1,000 LLM calls per problem).

I found the analyses of performance across equation complexity levels (Figure 4) and in-domain versus out-of-domain generalization (Figure 5) particularly informative. The correlation between symbolic accuracy and OOD performance (Figure 6) is an interesting finding that validates their evaluation approach.

However, it would be good to include more detailed case studies or error analyses to better understand which types of equations or mathematical patterns pose the greatest challenges for current methods.

**Methods And Evaluation Criteria:**

The methods for creating the benchmark are well-designed and appropriate. The LSR-Transform approach uses rigorous symbolic transformation while maintaining appropriate complexity and ensuring analytical solvability. The LSR-Synth methodology includes careful verification of both solvability (using numerical solvers) and scientific plausibility (through expert validation).

The evaluation metrics are comprehensive, including both numeric performance (accuracy to tolerance, NMSE) and symbolic accuracy. I particularly appreciate the validation of their GPT-4o-based symbolic evaluation against human experts (94.6% agreement), which strengthens confidence in their assessment approach. The inclusion of out-of-domain evaluation is also valuable for assessing true scientific understanding.

**Other Comments Or Suggestions:**

N/A

**Other Strengths And Weaknesses:**

Strengths:
- The benchmark addresses a gap in evaluating LLMs on scientific discovery vs. memorization
- The evaluation across multiple methods, backbones, and domains is thorough
- The correlation between symbolic accuracy and OOD performance is an interesting finding
- The examples of output hypotheses in Figure 14 provide good qualitative insights

Weaknesses:
- While showing that current methods struggle, there's limited analysis of why they fail or which specific reasoning capabilities are lacking
- The synthetic problems, while validated for plausibility, may not perfectly capture real scientific discovery challenges
- Could benefit from more discussion of how its findings might influence future LLM designs to improve scientific reasoning
- Would be nice to see more discussion around if some of the equation transformations in LSR-Transform, while mathematically valid, are meaningful from a scientific perspective

**Questions For Authors:**

1. Have you identified specific patterns or characteristics in the equations where current methods consistently fail?
2. How sensitive is performance on your benchmark to the quality or quantity of provided data? In real scientific scenarios, data is often limited or noisy - have you explored this dimension?
3. The correlation between symbolic accuracy and OOD performance is intriguing. Does this relationship hold equally across all scientific domains, or are there areas where symbolic understanding is less predictive of generalization?
4. Your benchmark focuses on discovering equations given data and context. Have you considered extending it to evaluate LLMs' abilities to generate plausible hypotheses before seeing complete datasets, which is another important aspect of scientific discovery?

**Relation To Broader Scientific Literature:**

The paper effectively places itself within the broader literature on symbolic regression, scientific equation discovery, and LLM reasoning capabilities. It builds upon previous benchmarks (SRBench, SRSD) while addressing their limitations for LLM evaluation. The connection to literature on LLM reasoning fragility with unfamiliar representations (Mirzadeh et al., 2024; Xie et al., 2024) provides solid theoretical grounding for their approach.

**Theoretical Claims:**

N/A.

---

> ### Author Rebuttal · Authors · 2025-04-01
>
> Thank you for dedicating your time and expertise to review our submission. Please find our responses below.
>
> > * However, it would be good to include more detailed case studies or error analyses to better understand which types of equations or mathematical patterns pose the greatest challenges for current methods.
> > * While showing that current methods struggle, there's limited analysis of why they fail or which specific reasoning capabilities are lacking
>
> Thank you for the thoughtful suggestion. We examined the failure cases during rebutal period and noticed that LLM-based models usually don't fail due to the syntax function errors (such as missing variables or dependencies). Instead, most errors come from incorrect terms or combination of terms within the equations, reflecting mistakes due to semantic aspects of discovery for a given scientific problem. We could not identify a uniform pattern of specific mathematical relations that consistently challenge current discovery methods. We believe this is because LLM-based equation discovery involves a complex interplay of domain knowledge, search capabilities with data-driven feedback, and mathematical manipulation skills. To thoroughly study failure case patterns would require investigating all these dimensions. We agree with the reviewer on the importance of this research direction and consider it a promising avenue for future work.
>
>
> > The correlation between symbolic accuracy and OOD performance is intriguing. Does this relationship hold equally across all scientific domains, or are there areas where symbolic understanding is less predictive of generalization?
>
> Thank you for this thoughtful question. We have actually explored the correlation between symbolic accuracy and OOD performance across different domains (detailed in Figure 13 Appendix). Our results demonstrate that this positive correlation holds across all four domains. Symbolic understanding appears to be a reliable predictor of generalization capability in equation discovery.
>
>
>
> > How sensitive is performance on your benchmark to the quality or quantity of provided data? In real scientific scenarios, data is often limited or noisy - have you explored this dimension?
>
> We have not yet explored the impact of noise or data limitations in our current benchmark generation process. Our primary motivation was to develop a benchmark that helps the community build better general LLM-based equation discovery agents for scientific domains, addressing limitations in existing benchmarks for emerging LLM-based techniques. We fully agree with the reviewer on the significance of noise and limited data in real-world scientific discovery scenarios. This is one of the directions we're considering for future benchmark enhancements.
>
>
> > Your benchmark focuses on discovering equations given data and context. Have you considered extending it to evaluate LLMs' abilities to generate plausible hypotheses before seeing complete datasets, which is another important aspect of scientific discovery?
>
> We would like to clarify that our benchmark evaluates SOTA methods that already incorporate a two-phase approach: hypothesis generation followed by data-driven validation. In the first phase, LLMs generate plausible equation hypotheses without seeing the data, which are then refined based on how well they fit the data. We agree that evaluating pure hypothesis generation capabilities is an important dimension of scientific discovery. While our present focus has been on end-to-end data-driven scientific equation discovery, extending the benchmark to explicitly measure hypothesis quality before the data validation is also a valuable direction for future work. We also think that combining mathematical derivation processes with data-driven reasoning represents an interesting avenue for future research that has not yet been thoroughly explored in current methods.
>
>
> > The examples of output hypotheses in Figure 14 provide good qualitative insights
>
> As some reviewers have higlighted this example helpful, we plan to include more hypothesis examples from different domains in the revision of paper.
>
>
> > * Could benefit from more discussion of how its findings might influence future LLM designs to improve scientific reasoning
> > * Would be nice to see more discussion around if some of the equation transformations in LSR-Transform, while mathematically valid, are meaningful from a scientific perspective
>
> Thank you for the suggestion. We will make sure to add more discussion regarding these points in the camera-ready version.

---

### Official Review · Reviewer_rJd1 · 2025-03-17

**Overall Recommendation:** 5

**Summary:**

This paper introduces LLM-SRBench, a benchmark designed to evaluate Large Language Models' capabilities in scientific equation discovery. The authors address a limitation in existing benchmarks: they primarily consist of well-known equations from textbooks that LLMs may have memorized during training, potentially leading to performance metrics that reflect recitation rather than discovery abilities.
LLM-SRBench comprises 239 problems across two categories:

LSR-Transform (111 problems): Transforms Feynman physics equations into alternative mathematical forms by changing which variable is solved for, challenging LLMs to discover less familiar representations of known physical relationships.
LSR-Synth (128 problems): Creates problems across chemistry, biology, physics, and material science by combining established scientific terms with synthetic terms, requiring models to employ both scientific reasoning and data-driven discovery.

The authors evaluate several LLM-based equation discovery methods (Direct Prompting, SGA, LaSR, LLM-SR) using various LLM backbones (Llama-3.1-8B, GPT-3.5-turbo, GPT-4o-mini). Their findings show that the best-performing method achieves 31.5% symbolic accuracy, indicating challenges in scientific equation discovery.
The paper also presents an evaluation methodology that considers both data fidelity (numeric accuracy) and symbolic correctness, including an LLM-based approach for assessing mathematical equivalence across different representations. The authors note a correlation between symbolic accuracy and out-of-domain generalization.
This work provides a testbed for current methods and may contribute to advancing scientific equation discovery research.

**Claims And Evidence:**

The paper convincingly asserts that LLM memorization is a factor in existing benchmarks, with memorization demonstrated through error curves; the authors leave open the possibility of alternative explanations.  They further demonstrate the benchmark's difficulty with a convincing performance evaluation across methods and SOTA-standard LLMs.

**Essential References Not Discussed:**

I am not aware of any essential references that are not discussed.

**Experimental Designs Or Analyses:**

The paper provides appropriately comprehensive coverage by standardizing methods to 1k LLM calls while preserving core algorithmic structures, evaluating multiple methods (Direct Prompting, SGA, LaSR, LLM-SR) across different LLM implementations. Their comparison of performance across equivalent complexity levels demonstrates that challenge stems from semantic transformation rather than structural complexity. The correlation analysis between symbolic accuracy and OOD performance validates their evaluation metrics. However, the design lacks ablation studies to isolate which components drive performance differences, and while domain performance variations are noted, there's no systematic exploration of whether different methods have domain-specific advantages.

**Methods And Evaluation Criteria:**

The paper utilizes a two-pronged approach for dataset creation: transforming known equations through variable substitution and combining known scientific terms with synthetic elements. The benchmark spans four scientific domains with varying distribution of problems across domains. The evaluation employs dual metrics focusing on data fidelity (numeric accuracy/error) and symbolic accuracy, with separate assessments for in-domain and out-of-domain generalization. The authors introduce an LLM-based approach for evaluating symbolic equivalence, validated through comparison with human judgments. The paper evaluates various LLM-based scientific equation discovery methods but does not include non-LLM symbolic regression baselines for comparison. Questions remain about whether the synthetic problems authentically model scientific discovery processes rather than testing mathematical manipulation skills.

**Other Comments Or Suggestions:**

None.

**Other Strengths And Weaknesses:**

This paper is very well written and seems to explain the nuances of the field well, deserving to be a part of the published record.

**Questions For Authors:**

Do you have either intuition or  insight into specific criteria for when numeric performance and symbolic accuracy metrics diverge?

The benchmark design assumes that transforming equations (LSR-Transform) and combining known with synthetic terms (LSR-Synth) effectively models scientific discovery. How do you justify that these approaches genuinely reflect how scientists discover new equations in practice, rather than simply creating mathematically challenging problems? Would performance on these tasks predict success in real scientific discovery?

**Relation To Broader Scientific Literature:**

While I am not an expert in this field, the paper appears to effectively situate itself within several research domains, connecting to existing symbolic regression benchmarks while addressing their limitations for LLM evaluation

**Theoretical Claims:**

This paper is empirical in nature.  No significant theoretical claims are made.

---

> ### Author Rebuttal · Authors · 2025-04-01
>
> Thank you for dedicating your time and expertise to review our submission. Please find our responses below.
>
>
> > * there's no systematic exploration of whether different methods have domain-specific advantages.
>
> We agree this is an important consideration, but it falls outside our study's scope. Our benchmark aims to help community towards building better general LLM-based equation discovery agents applicable across domains. Domain-specific variations mostly come from the LLM's knowledge rather than the agentic discovery framework itself. Further experiments with different LLM backbones would be helpful to study this question in more depth.
>
>
> > Do you have either intuition or insight into specific criteria for when numeric performance and symbolic accuracy metrics diverge?
>
>
> Thank you for the thoughtful question. This is indeed a common challenge in equation discovery. Equations with good numeric accuracy may differ significantly in symbolic manner. Conversely, equations with similar symbolic structures can also exhibit significantly different numeric behaviors due to constant/coefficient variations affecting function behavior. This is why our benchmark incorporates both numeric and symbolic metrics for comprehensive evaluation.
>
>
>
> > The benchmark design assumes that transforming equations (LSR-Transform) and combining known with synthetic terms (LSR-Synth) effectively models scientific discovery. How do you justify that these approaches genuinely reflect how scientists discover new equations in practice, rather than simply creating mathematically challenging problems? Would performance on these tasks predict success in real scientific discovery?
>
> Thanks for raising this important question. Whether performance on these tasks predicts success in real scientific discovery remains an open research question. But we think that our benchmark can be a good starting point to guide models towards that direction, particularly in the task of scientific equation discovery.
> To ensure our benchmark reflects scientific discovery rather than just creating mathematically challenging problems, we implemented filtering steps in the design of benchmark (detailed in Section 2, Figures 4, 8, and 10) to avoid lengthy problems with excessive mathematical and syntax complexity.

---

### Decision · Program_Chairs · 2025-05-01

**Decision:**

Accept (oral)

**Comment:**

**(a) Summary**
This paper presents LLM-SRBench, a new benchmark designed to rigorously assess scientific equation discovery capabilities of large language models (LLMs) while mitigating memorization effects. It introduces two components: (1) LSR-Transform, which reformulates familiar physics equations to challenge recall-based strategies; and (2) LSR-Synth, which introduces synthetic equations that require novel hypothesis generation across four scientific domains. The benchmark evaluates both symbolic and numeric accuracy and highlights current limitations in LLMs' generalization and symbolic reasoning. The best-performing method achieves only 31.5% symbolic accuracy, underscoring the difficulty of the task and motivating further work in this direction.

**(b) Strengths**
The reviewers agree on several key strengths of the paper:
* Novel benchmark design:
rJd1, Yany, and vLgS commend the benchmark’s effort to eliminate memorization by introducing reformulated and synthetic equations.
* Thorough evaluation:
Yany and vLgS highlight the comprehensive experiments using multiple LLMs (GPT-4o-mini, LLaMA, GPT-3.5) and methods (LLM-SR, LaSR, SGA, etc.).
* Sound methodology:
All reviewers appreciate the dual evaluation approach using symbolic accuracy and numeric fidelity. Yany and vLgS find the use of GPT-4o for symbolic equivalence evaluation well-justified and empirically validated.
* Contribution to broader research:
tSk5 and rJd1 praise the benchmark’s relevance to advancing scientific discovery capabilities in LLMs and symbolic regression.

**(c) Weaknesses**
Despite the paper’s merits, reviewers identified several areas for improvement:
* Evaluation Limitations:
tSk5 questions the sole reliance on GPT-4o for symbolic similarity evaluation and suggests including more classical evaluation techniques (e.g., R²), though the authors rebut this with empirical justification.
* Novelty validation:
vLgS raises concerns about GPT-4o being the only novelty evaluator in LSR-Synth, recommending secondary human or literature-based verification.
* Lack of failure analysis:
vLgS and Yany request deeper insight into failure patterns—such as domain-specific trends, frequent error types, and case studies—though the authors plan to include this in the camera-ready version.
* Generalization gaps:
vLgS and Yany observe that performance disparities across domains (e.g., chemistry vs. physics) lack sufficient causal analysis, though complexity statistics were later provided.
* Missing baselines:
tSk5 recommends comparing with traditional symbolic regression methods. The authors address this by adding PySR results showing poor symbolic accuracy despite good numeric fit, strengthening the LLM motivation.
* Data realism:
tSk5 also notes the benchmark could benefit from incorporating noise or real-world data to better reflect experimental uncertainty, which the authors acknowledge as future work.

**(d) Decision**
Overall, the reviewers are positive about the paper, recognizing its timely contribution and high-quality benchmark design. While several concerns were raised, the authors addressed them constructively during the rebuttal, adding ablations, baselines (e.g., PySR), and deeper analyses. Remaining weaknesses, such as GPT-based novelty evaluation and limited failure breakdowns, are acknowledged by the authors as areas for future development. Given its novelty, empirical rigor, and relevance to the growing field of LLM-driven scientific reasoning, I recommend acceptance.

**Additional Comments on Reviewer Discussion**
* rJd1: Read the rebuttal, appreciated the authors’ clarifications, and maintained a positive stance (Strong Accept).
* Yany: Engaged with the rebuttal and acknowledged the authors’ responses, especially around symbolic accuracy and domain-wise correlations.
* vLgS: Initially raised significant concerns about novelty and domain differences, but found the rebuttal clarifying and helpful, though GPT-based novelty checks remain a lingering issue.
* tSk5: Provided constructive evaluation-focused suggestions; acknowledged the rebuttal and appreciated the added comparisons with classical SR methods.